# Likelihood approximation networks (LANs) for fast inference of simulation models in cognitive neuroscience

Alexander Fengler[1,2]*, Lakshmi N Govindarajan[1,2], Tony Chen[3], Michael J Frank[1,2]*

[1]Department of Cognitive, Linguistic and Psychological Sciences, Brown University, Providence, United States; [2]Carney Institute for Brain Science, Brown University, Providence, United States; [3]Psychology and Neuroscience Department, Boston College, Chestnut Hill, United States

**Abstract** In cognitive neuroscience, computational modeling can formally adjudicate between theories and affords quantitative fits to behavioral/brain data. Pragmatically, however, the space of plausible generative models considered is dramatically limited by the set of models with known likelihood functions. For many models, the lack of a closed-form likelihood typically impedes Bayesian inference methods. As a result, standard models are evaluated for convenience, even when other models might be superior. Likelihood-free methods exist but are limited by their computational cost or their restriction to particular inference scenarios. Here, we propose neural networks that learn approximate likelihoods for arbitrary generative models, allowing fast posterior sampling with only a one-off cost for model simulations that is amortized for future inference. We show that these methods can accurately recover posterior parameter distributions for a variety of neurocognitive process models. We provide code allowing users to deploy these methods for arbitrary hierarchical model instantiations without further training.

**\*For correspondence:**
alexander_fengler@brown.edu (AF);
Michael_Frank@brown.edu (MJF)

## Introduction

Computational modeling has gained traction in cognitive neuroscience in part because it can guide principled interpretations of functional demands of cognitive systems while maintaining a level of tractability in the production of quantitative fits of brain-behavior relationships. For example, models of reinforcement learning (RL) are frequently used to estimate the neural correlates of the exploration/exploitation tradeoff, of asymmetric learning from positive versus negative outcomes, or of model-based vs. model-free control (*Schönberg et al., 2007*; *Niv et al., 2012*; *Frank et al., 2007*; *Zajkowski et al., 2017*; *Badre et al., 2012*; *Daw et al., 2011b*). Similarly, models of dynamic decision-making processes are commonly used to disentangle the strength of the evidence for a given choice from the amount of that evidence needed to commit to any choice, and how such parameters are impacted by reward, attention, and neural variability across species (*Rangel et al., 2008*; *Forstmann et al., 2010*; *Krajbich and Rangel, 2011*; *Frank et al., 2015*; *Yartsev et al., 2018*; *Doi et al., 2020*). Parameter estimates might also be used as a theoretically driven method to reduce the dimensionality of brain/behavioral data that can be used for prediction of, for example, clinical status in computational psychiatry (*Huys et al., 2016*).

Interpreting such parameter estimates requires robust methods that can estimate their generative values, ideally including their uncertainty. For this purpose, Bayesian statistical methods have gained traction. The basic conceptual idea in Bayesian statistics is to treat parameters $\theta$ and data $\mathbf{x}$ as stemming from a joint probability model $p(\theta, \mathbf{x})$. Statistical inference proceeds by using Bayes' rule,

**eLife digest** Cognitive neuroscience studies the links between the physical brain and cognition. Computational models that attempt to describe the relationships between the brain and specific behaviours quantitatively is becoming increasingly popular in this field. This approach may help determine the causes of certain behaviours and make predictions about what behaviours will be triggered by specific changes in the brain.

Many of the computational models used in cognitive neuroscience are built based on experimental data. A good model can predict the results of new experiments given a specific set of conditions with few parameters. Candidate models are often called 'generative': models that can simulate data. However, typically, cognitive neuroscience studies require going the other way around: they need to infer models and their parameters from experimental data. Ideally, it should also be possible to properly assess the remaining uncertainty over the parameters after having access to the experimental data. To facilitate this, the Bayesian approach to statistical analysis has become popular in cognitive neuroscience.

Common software tools for Bayesian estimation require a 'likelihood function', which measures how well a model fits experimental data for given values of the unknown parameters. A major obstacle is that for all but the most common models, obtaining precise likelihood functions is computationally costly. In practice, this requirement limits researchers to evaluating and comparing a small subset of neurocognitive models for which a likelihood function is known. As a result, it is convenience, rather than theoretical interest, that guides this process.

In order to provide one solution for this problem, Fengler et al. developed a method that allows users to perform Bayesian inference on a larger number of models without high simulation costs. This method uses likelihood approximation networks (LANs), a computational tool that can estimate likelihood functions for a broad class of models of decision making, allowing researchers to estimate parameters and to measure how well models fit the data. Additionally, Fengler et al. provide both the code needed to build networks using their approach and a tutorial for users.

The new method, along with the user-friendly tool, may help to discover more realistic brain dynamics underlying cognition and behaviour as well as alterations in brain function.

$$p(\theta|\mathbf{x}) = \frac{p(\mathbf{x}|\theta)p(\theta)}{p(\mathbf{x})}$$

to get at $p(\theta|\mathbf{x})$, the posterior distribution over parameters' given data. $p(\theta)$ is known as the prior distribution over the parameters θ, and $p(\mathbf{x})$ is known as the evidence or just probability of the data (a quantity of importance for model comparison). The term $p(\mathbf{x}|\theta)$, the probability (density) of the dataset $\mathbf{x}$ given parameters θ, is known as the likelihood (in accordance with usual notation, we will also write $\ell(\theta|\mathbf{x})$ in the following). It is common in cognitive science to represent likelihoods as $p(\mathbf{x}|\theta, s)$, where $s$ specifies a particular stimulus. We suppress $s$ in our notation, but note that our approach easily generalizes when explicitly conditioning on trial-wise stimuli. Bayesian parameter estimation is a natural way to characterize uncertainty over parameter values. In turn, it provides a way to identify and probe parameter tradeoffs. While we often do not have access to $p(\theta|\mathbf{x})$ directly, we can draw samples from it instead, for example, via Markov Chain Monte Carlo (MCMC) (*Robert and Casella, 2013*; *Diaconis, 2009*; *Robert and Casella, 2011*).

Bayesian estimation of the full posterior distributions over model parameters contrasts with maximum likelihood estimation (MLE) methods that are often used to extract single best parameter values, without considering their uncertainty or whether other parameter estimates might give similar fits. Bayesian methods naturally extend to settings that assume an implicit hierarchy in the generative model in which parameter estimates at the individual level are informed by the distribution across a group, or even to assess within an individual how trial-by-trial variation in (for example) neural activity can impact parameter estimates (commonly known simply as hierarchical inference). Several toolboxes exist for estimating the parameters of popular models like the drift diffusion model (DDM) of decision-making and are widely used by the community for this purpose (*Wiecki et al., 2013*; *Heathcote et al., 2019*; *Turner et al., 2015*; *Ahn et al., 2017*). Various studies have used

these methods to characterize how variability in neural activity, and manipulations thereof, alters learning and decision parameters that can quantitatively explain variability in choice and response time distributions (*Cavanagh et al., 2011*; *Frank et al., 2015*; *Herz et al., 2016*; *Pedersen and Frank, 2020*).

Traditionally, however, posterior sampling or MLE for such models required analytical likelihood functions: a closed-form mathematical expression providing the likelihood of observing specific data (reaction times and choices) for a given model and parameter setting. This requirement limits the application of any likelihood-based method to a relatively small subset of cognitive models chosen for so-defined convenience instead of theoretical interest. Consequently, model comparison and estimation exercises are constrained as many important but likelihood-free models were effectively *untestable* or required weeks to process a single model formulation. Testing any slight adjustment to the generative model (e.g., different hierarchical grouping or splitting conditions) requires a repeated time investment of the same order. For illustration, we focus on the class of sequential sampling models (SSMs) applied to decision-making scenarios, with the most common variants of the DDM. The approach is, however, applicable to any arbitrary domain.

In the standard DDM, a two-alternative choice decision is modeled as a noisy accumulation of evidence toward one of two decision boundaries (*Ratcliff and McKoon, 2008*). This model is widely used as it can flexibly capture variations in choice, error rates, and response time distributions across a range of cognitive domains and its parameters have both psychological and neural implications. While the likelihood function is available for the standard DDM and some variants including inter-trial variability of its drift parameter, even seemingly small changes to the model form, such as dynamically varying decision bounds (*Cisek et al., 2009*; *Hawkins et al., 2015*) or multiple-choice alternatives (*Krajbich and Rangel, 2011*), are prohibitive for likelihood-based estimation, and instead require expensive Monte Carlo (MC) simulations, often without providing estimates of uncertainty across parameters.

In the last decade and a half, approximate Bayesian computation (ABC) algorithms have grown to prominence (*Sisson et al., 2018*). These algorithms enable one to sample from posterior distributions over model parameters, where models are defined only as simulators, which can be used to construct empirical likelihood distributions (*Sisson et al., 2018*). ABC approaches have enjoyed successful application across life and physical sciences (e.g., *Akeret et al., 2015*), and notably, in cognitive science (*Turner and Van Zandt, 2018*), enabling researchers to estimate theoretically interesting models that were heretofore intractable. However, while there have been many advances without sacrificing information loss in the posterior distributions (*Turner and Sederberg, 2014*; *Holmes, 2015*), such ABC methods typically require many simulations to generate synthetic or empirical likelihood distributions, and hence can be computationally expensive (in some cases prohibitive – it can take weeks to estimate parameters for a single model). This issue is further exacerbated when embedded within a sequential MCMC sampling scheme, which is needed for unbiased estimates of posterior distributions. For example, one typically needs to simulate between 10,000 and 100,000 times (the exact number varies depending on the model) for each proposed combination of parameters (i.e., for each sample along a Markov chain [MC], which may itself contain tens of thousands of samples), after which they are discarded. This situation is illustrated in *Figure 1*, where the red arrows point at the computations involved in the approach suggested by *Turner et al., 2015*.

To address this type of issue, the statistics and machine learning communities have increasingly focused on strategies for the amortization of simulation-based computations (*Gutmann et al., 2018*; *Papamakarios and Murray, 2016*; *Papamakarios et al., 2019a*; *Lueckmann et al., 2019*; *Radev et al., 2020b*; *Radev et al., 2020a*; *Gonçalves et al., 2020*; *Järvenpää et al., 2021*). The aim is generally to use model simulations upfront and learn a reusable approximation of the function of interest (targets can be the likelihood, the evidence, or the posterior directly).

In this paper, we develop a general ABC method (and toolbox) that allows users to perform inference on a significant number of neurocognitive models without repeatedly incurring substantial simulation costs. To motivate our particular approach and situate it in the context of other methods, we outline the following key desiderata:

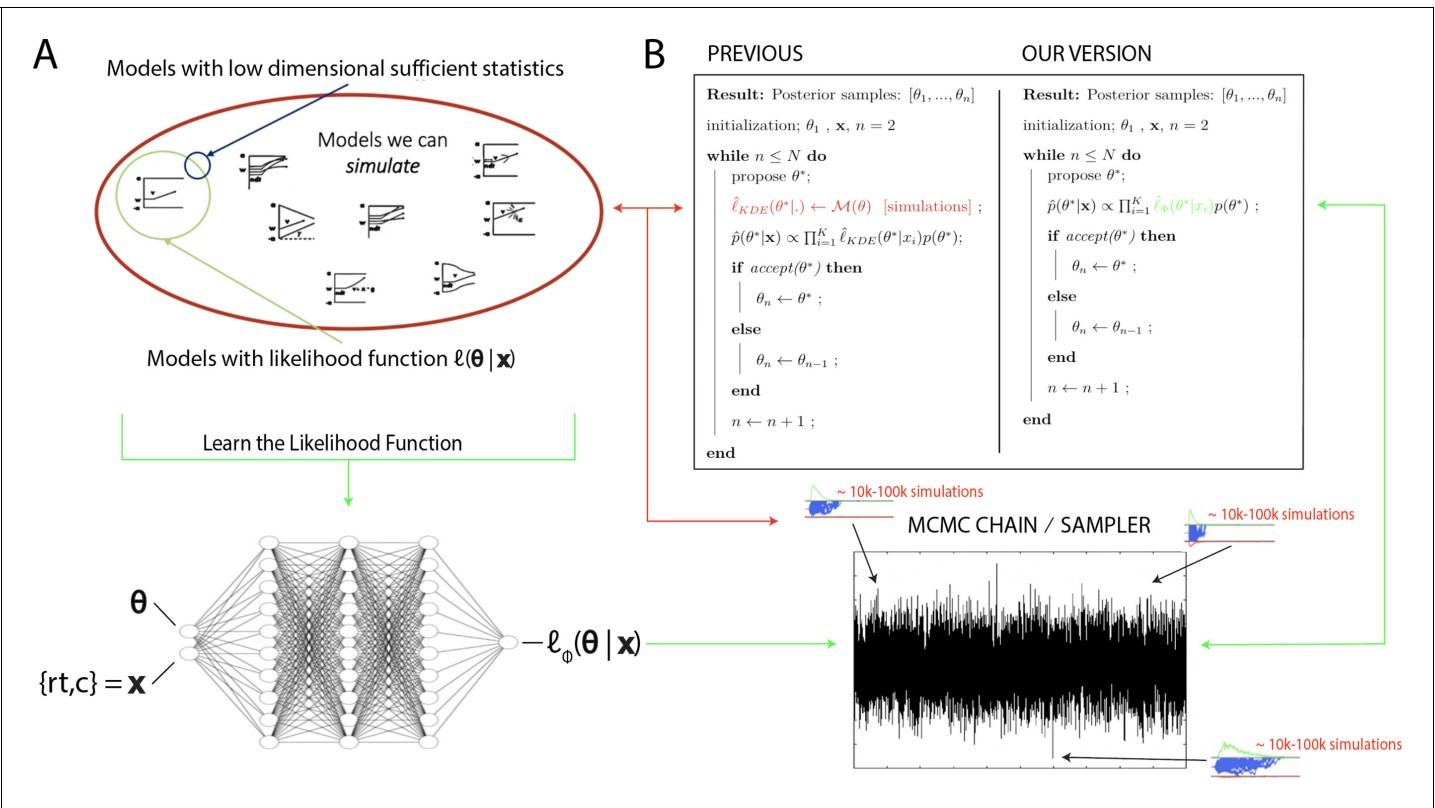

**Figure 1.** High level overview of the proposed methods. (**A**) The space of theoretically interesting models in the cognitive neurosciences (red) is much larger than the space of mechanistic models with analytical likelihood functions (green). Traditional approximate Bayesian computation (ABC) methods require models that have low-dimensional sufficient statistics (blue). (**B**) illustrates how likelihood approximation networks can be used in lieu of online simulations for efficient posterior sampling. The left panel shows the predominant 'probability density approximation' (PDA) method used for ABC in the cognitive sciences (*Turner et al., 2015*). For each step along a Markov chain, 10K–100K simulations are required to obtain a single likelihood estimate. The right panel shows how we can avoid the simulation steps during inference using amortized likelihood networks that have been pretrained using empirical likelihood functions (operationally in this paper: kernel density estimates and discretized histograms).

1. The method needs to be easily and rapidly deployable to end users for Bayesian inference on various models hitherto deemed computationally intractable. This desideratum naturally leads us to an amortization approach, where end users can benefit from costs incurred upfront.
2. Our method should be sufficiently flexible to support arbitrary inference scenarios, including hierarchical inference, estimation of latent covariate (e.g., neural) processes on the model parameters, arbitrary specification of parameters across experimental conditions, and without limitations on dataset sizes. This desideratum leads us to amortize the likelihood functions, which (unlike other amortization strategies) can be immediately applied to arbitrary inference scenarios without further cost.
3. We desire approximations that do not a priori sacrifice covariance structure in the parameter posteriors, a limitation often induced for tractability in variational approaches to approximate inference (*Blei et al., 2017*).
4. End users should have access to a convenient interface that integrates the new methods seamlessly into preexisting workflows. The aim is to allow users to get access to a growing database of amortized models through this toolbox and enable increasingly complex models to be fit to experimental data routinely, with minimal adjustments to the user's working code. For this purpose, we will provide an extension to the widely used HDDM toolbox (*Wiecki et al., 2013*; *Pedersen and Frank, 2020*) for parameter estimation of DDM and RL models.
5. Our approach should exploit modern computing architectures, specifically parallel computation. This leads us to focus on the encapsulation of likelihoods into neural network (NN) architectures, which will allow batch processing for posterior inference.

Guided by these desiderata, we developed two amortization strategies based on NNs and empirical likelihood functions. Rather than simulate during inference, we instead train NNs as parametric function approximators to learn the likelihood function from an initial set of a priori simulations across a wide range of parameters. By learning (log) likelihood functions directly, we avoid posterior distortions that result from inappropriately chosen (user-defined) summary statistics and corresponding distance measures as applied in traditional ABC methods (*Sisson et al., 2018*). Once trained, likelihood evaluation only requires a forward pass through the NN (as if it were an analytical likelihood) instead of necessitating simulations. Moreover, any algorithm can be used to facilitate posterior sampling, MLE or maximum a posteriori estimation (MAP).

For generality, and because they each have their advantages, we use two classes of architectures, multilayered perceptrons (MLPs) and convolutional neural networks (CNNs), and two different posterior sampling methods (MCMC and importance sampling). We show proofs of concepts using posterior sampling and parameter recovery studies for a range of cognitive process models of interest. The trained NNs provide the community with a (continually expandable) bank of encapsulated likelihood functions that facilitate consideration of a larger (previously computationally inaccessible) set of cognitive models, with orders of magnitude speedup relative to simulation-based methods. This speedup is possible because costly simulations only have to be run once per model upfront and henceforth be avoided during inference: previously executed computations are amortized and then shared with the community.

Moreover, we develop standardized amortization pipelines that allow the user to apply this method to arbitrary models, requiring them to provide only a functioning simulator of their model of choice.

In the 'Approximate Bayesian Computation' section, we situate our approach in the greater context of ABC, with a brief review of online and amortization algorithms. The subsequent sections describe our amortization pipelines, including two distinct strategies, as well as their suggested use cases. The 'Test beds' section provides an overview of the (neuro)cognitive process models that comprise our benchmarks. The 'Results' section shows proof-of-concept parameter recovery studies for the two proposed algorithms, demonstrating that the method accurately recovers both the posterior mean and variance (uncertainty) of generative model parameters, and that it does so at a run-time speed of orders of magnitude faster than traditional ABC approaches without further training. We further demonstrate an application to hierarchical inference, in which our trained networks can be imported into widely used toolboxes for arbitrary inference scenarios. In the 'Discussion' and the last section, we further situate our work in the context of other ABC amortization strategies and discuss the limitations and future work.

## Approximate Bayesian computation

ABC methods apply when one has access to a parametric stochastic simulator (also referred to as generative model), but, unlike the usual setting for statistical inference, no access to an explicit mathematical formula for the likelihood of observations given the simulator's parameter setting.

While likelihood functions for a stochastic stimulator can in principle be mathematically derived, this can be exceedingly challenging even for some historically famous models such as the Ornstein–Uhlenbeck (OU) process (*Lipton and Kaushansky, 2018*) and may be intractable in many others. Consequently, the statistical community has increasingly developed ABC tools that enable posterior inference of such 'likelihood-free' stochastic models while completely bypassing any likelihood derivations (*Cranmer et al., 2020*).

Given a parametric stochastic simulator model $\mathcal{M}$ and dataset $\mathbf{x}$, instead of exact inference based on $p_{\mathcal{M}}(\theta|\mathbf{x})$, these methods attempt to draw samples from an approximate posterior $\tilde{p}_{\mathcal{M}}(\theta|\mathbf{x})$. Consider the following general equation:

$$\tilde{p}_{\mathcal{M}}(\theta|\mathbf{x}) \propto \int K_h(||s_{\mathcal{M}} - s_{\mathbf{x}}||) p_{\mathcal{M}}(s_{\mathcal{M}}|\theta) \, ds_{\mathcal{M}} \, \pi(\theta) \propto \tilde{p}_{\mathcal{M}}(\mathbf{x}|\theta) \pi(\theta)$$

where $s_{\mathcal{M}}$ refers to sufficient statistics (roughly, summary statistics of a dataset that retain sufficient information about the parameters of the generative process). $K_h(||s_{\mathcal{M}} - s_{\mathbf{x}}||)$ refers to a kernel-based distance measure/cost function, which evaluates a probability density function for a given distance between the observed and expected summary statistics $||s_{\mathcal{M}} - s_{\mathbf{x}}||$. The parameter $h$ (commonly

known as bandwidth parameter) modulates the cost gradient. Higher values of $h$ lead to more graceful decreases in cost (and therefore a worse approximation of the true posterior).

By generating simulations, one can use such summary statistics to obtain approximate likelihood functions, denoted as $\tilde{p}_{\mathcal{M}}(\mathbf{x}|\theta)$, where approximation error can be mitigated by generating large numbers of simulations. The caveat is that the amount of simulation runs needed to achieve a desired degree of accuracy in the posterior can render such techniques computationally infeasible.

With a focus on amortization, our goal is to leverage some of the insights and developments in the ABC community to develop NN architectures that can learn approximate likelihoods deployable for any inference scenario (and indeed any inference method, including MCMC, variational inference, or even MLE) without necessitating repeated training. We next describe our particular approach and return to a more detailed comparison to existing methods in the 'Discussion' section.

## Learning the likelihood with simple NN architectures

In this section, we outline our approach to amortization of computational costs of large numbers of simulations required by traditional ABC methods. Amortization approaches incur a one-off (potentially quite large) simulation cost to enable cheap, repeated inference for any dataset. Recent research has led to substantial developments in this area (*Cranmer et al., 2020*). The most straightforward approach is to simply simulate large amounts of data and compile a database of how model parameters are related to observed summary statistics of the data (*Mestdagh et al., 2019*). This database can then be used during parameter estimation in empirical datasets using a combination of nearest-neighbor search and local interpolation methods. However, this approach suffers from the curse of dimensionality with respect to storage demands (a problem that is magnified with increasing model parameters). Moreover, its reliance on summary statistics (*Sisson et al., 2018*) does not naturally facilitate flexible reuse across inference scenarios (e.g., hierarchical models, multiple conditions while fixing some parameters across conditions, etc.).

To fulfill all desiderata outlined in the introduction, we focus on directly encapsulating the likelihood function over empirical observations of a simulation model so that likelihood evaluation is substantially cheaper than constructing (empirical) likelihoods via model simulation online during inference. Such empirical observations could be, for example, trial-wise choices and reaction times (RTs). This strategy then allows for flexible reuse of such approximate likelihood functions $\hat{\ell}(\theta|\mathbf{x})$ in a large variety of inference scenarios applicable to common experimental design paradigms. Specifically, we encapsulate $\hat{\ell}(\theta|\mathbf{x})$ as a feed-forward NN, which allows for parallel evaluation by design. We refer to these networks as likelihood approximation networks (LANs).

*Figure 1* spells out the setting (A) and usage (B) of such a method. The LAN architectures used in this paper are simple, small in size, and are made available for download for local usage. While this approach does not allow for instantaneous posterior inference, it does considerably reduce computation time (by up to three orders of magnitude; see 'Run time' section) when compared to approaches that demand simulations at inference. Notably, this approach also substantially speeds up inference even for models that are not entirely likelihood free but nevertheless require costly numerical methods to obtain likelihoods. Examples are the full-DDM with inter-trial variability in parameters (for which likelihoods can be obtained via numerical integration, as is commonly done in software packages such as HDDM but with substantial cost), but also other numerical methods for generalized DDMs (*Shinn et al., 2020*; *Drugowitsch, 2016*). At the same time, we maintain the high degree of flexibility with regards to deployment across arbitrary inference scenarios. As such, LANs can be treated as a highly flexible plug-in to existing inference algorithms and remain conceptually simple and lightweight.

Before elaborating our LAN approach, we briefly situate it in the context of some related work.

One branch of literature that interfaces ABC with deep learning attempts to amortize posterior inference directly in end-to-end NNs (*Radev et al., 2020b*; *Radev et al., 2020a*; *Papamakarios and Murray, 2016*; *Papamakarios et al., 2019a*; *Gonçalves et al., 2020*). Here, NN architectures are trained with a large number of simulated datasets to produce posterior distributions over parameters, and once trained, such networks can be applied to directly estimate parameters from new datasets without the need for further simulations. However, the goal to directly estimate posterior parameters from data requires the user to first train a NN for the very specific inference scenario in which it is applied empirically. Such approaches are not easily deployable if a user wants to test

multiple inference scenarios (e.g., parameters may vary as a function of task condition or brain activity, or in hierarchical frameworks), and consequently, they do not achieve our second desideratum needed for a user-friendly toolbox, making inference cheap and simple.

We return to discuss the relative merits and limitations of these and further alternative approaches in the 'Discussion' section.

Formally, we use model simulations to learn a function $f_\Phi(x, \theta)$, where $f_\Phi(.)$ is the output of a NN with parameter vector $\Phi$ (weights, biases). The function $f_\Phi(\mathbf{x}, \theta)$ is used as an approximation $\hat{\ell}(\theta|\mathbf{x})$ of the likelihood function $\ell(\theta|\mathbf{x})$. Once learned, we can use such $f$ as a plug-in to Bayes' rule:

$$\hat{p}(\theta|\mathbf{x}) \propto \prod_{i=1}^{N} \hat{\ell}(\theta|x_i)p(\theta)$$

To perform posterior inference, we can now use a broad range of existing MC or MCMC algorithms. We note that, fulfilling our second desideratum, an extension to hierarchical inference is as simple as plugging in our NN into a probabilistic model of the form

$$\hat{p}(\theta|\mathbf{x}) \propto \prod_{j=1}^{J}\prod_{i=1}^{N} \hat{\ell}(\theta_j|x_{ji})p(\theta_j|\alpha,\beta)p(\alpha,\beta|\gamma)$$

where $\alpha$, $\beta$ are generic group-level global parameters, and $\gamma$ serves as a generic fixed hyperparameter vector.

We provide proofs of concepts for two types of LANs. While we use MLPs and convolutional neural networks (CNNs), of conceptual importance is the distinction between the two problem representations they tackle, rather than the network architectures per se.

The first problem representation, which we call the pointwise approach, considers the functions $f_\Phi(x|\theta)$, where $\theta$ is the parameter vector of a given stochastic simulator model, and $x$ is a single datapoint (trial outcome). The pointwise approach is a mapping from the input dimension $|\Theta| + |x|$, where $|.|$ refers to the cardinality, to the one-dimensional output. The output is simply the log-likelihood of the single datapoint $x$ given the parameter vector $\theta$. As explained in the next section, for this mapping we chose simple MLPs.

The second problem representation, which we will refer to as the histogram approach, instead aims to learn a function $f_\Phi(.|\theta)$, which maps a parameter vector $\theta$ to the likelihood over the full (discretized) dataspace (i.e., the likelihood of the entire RT distributions at once). We represent the output space as an outcome histogram with dimensions $n \times m$ (where in our applications $n$ is the number of bins for a discretization of reaction times, and $m$ refers to the number of distinct choices). Thus, our mapping has input dimension $|\Theta|$ and output dimension $|m \times n|$. Representing the problem this way, we chose CNNs as the network architecture.

In each of the above cases, we pursued the architectures that seemed to follow naturally, without any commitment to their optimality. The pointwise approach operates on low-dimensional inputs and outputs. With network evaluation speed being of primary importance to us, we chose a relatively shallow MLP to learn this mapping, given that it was expressive enough. However, when learning a mapping from a parameter vector $\theta$ to the likelihood over the full dataspace, as in the histogram approach, we map a low-dimensional data manifold to a much higher dimensional one. Using an MLP for this purpose would imply that the number of NN parameters needed to learn this function would be orders of magnitude larger than using a CNN. Not only would this mean that forward passes through the network would take longer, but also an increased propensity to overfit on an identical data budget.

The next two sections will give some detail regarding the networks chosen for the pointwise and histogram LANs. *Figure 1* illustrates the general idea of our approach while *Figure 2* gives a conceptual overview of the exact training and inference procedure proposed. These details are thoroughly discussed in the last section of the paper.

## Pointwise approach: learn likelihoods of individual observations with MLPs

As a first approach, we use simple MLPs to learn the likelihood function of a given stochastic simulator. The network learns the mapping $\{\theta, \mathbf{x}\} \rightarrow \log \ell_\Phi(\theta|\mathbf{x})$, where $\theta$ represents the parameters of our

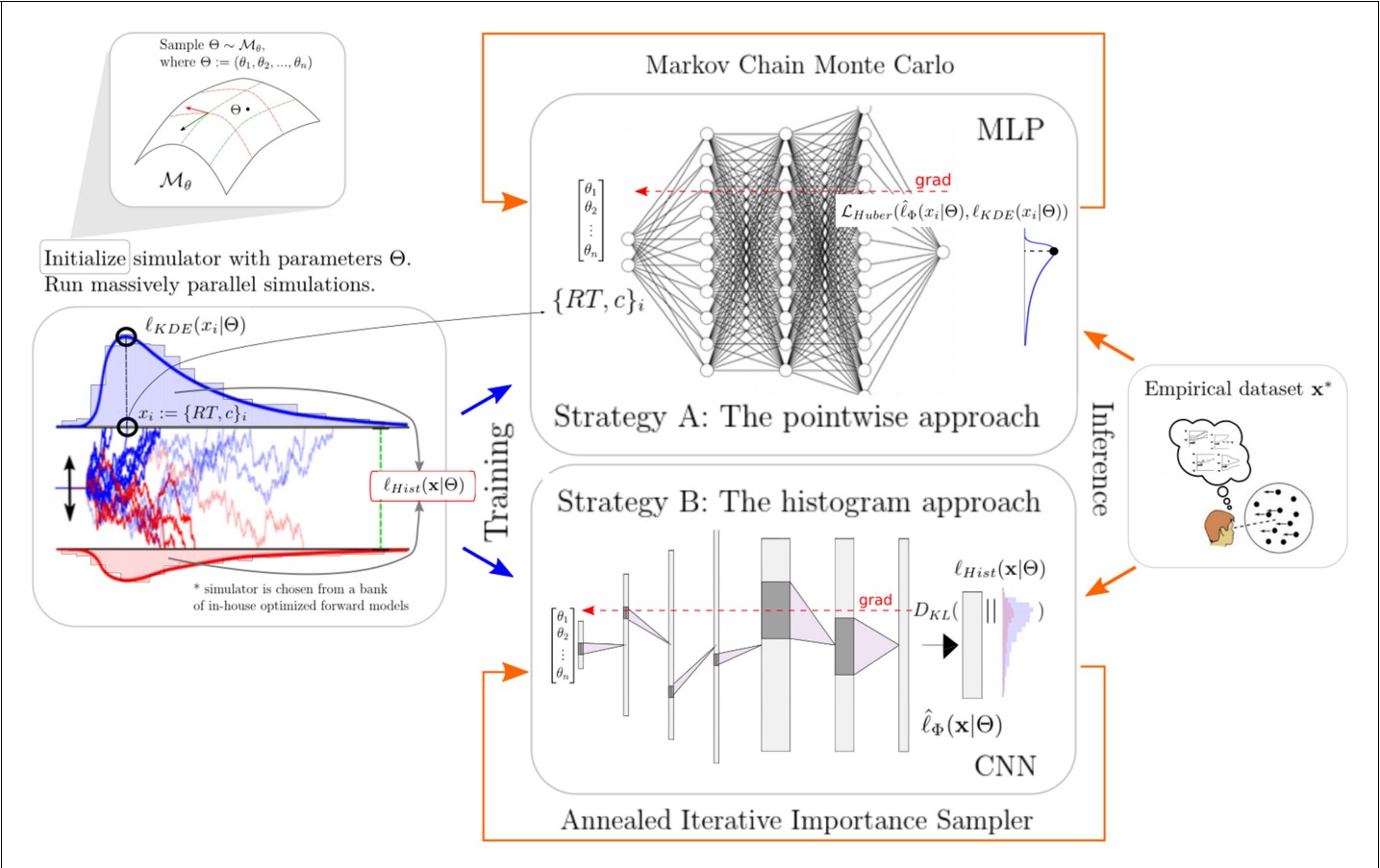

**Figure 2.** High-level overview of our approaches. For a given model $\mathcal{M}$, we sample model parameters θ from a region of interest (left 1) and run 100k simulations (left 2). We use those simulations to construct a kernel density estimate-based empirical likelihood, and a discretized (histogram-like) empirical likelihood. The combination of parameters and the respective likelihoods is then used to train the likelihood networks (right 1). Once trained, we can use the multilayered perceptron and convolutional neural network for posterior inference given an empirical/experimental dataset (right 2).

stochastic simulator, $\mathbf{x}$ are the simulator outcomes (in our specific examples below, $\mathbf{x}$, refers to the tuple $(rt, c)$ of reaction times and choices), and $\log \ell(\theta|\mathbf{x})$ is the log-likelihood of the respective outcome. The trained network then serves as the approximate likelihood function $\log \hat{\ell}_\Phi(\theta|\mathbf{x})$. We emphasize that the network is trained as a function approximator to provide us with a computationally cheap likelihood estimate, not to provide a surrogate simulator. Biological plausibility of the network is therefore not a concern when it comes to architecture selection. Fundamentally this approach attempts to learn the log-likelihood via a nonlinear regression, for which we chose an MLP. Log-likelihood labels for training were derived from empirical likelihoods functions (details are given in the 'Materials and methods' section), which in turn were constructed as kernel density estimates (KDEs). The construction of KDEs roughly followed *Turner et al., 2015*. We chose the Huber loss function (details are given in the 'Materials and methods' section) because it is more robust to outliers and thus less susceptible to distortions that can arise in the tails of distributions.

## Histogram approach: learn likelihoods of entire dataset distributions with CNNs

Our second approach is based on a CNN architecture. Whereas the MLP learned to output a single scalar likelihood output for each datapoint ('trial', given a choice, reaction time, and parameter vector), the goal of the CNN was to evaluate, for the given model parameters, the likelihood of an arbitrary number of datapoints via one forward pass through the network. To do so, the output of the CNN was trained to produce a probability distribution over a discretized version of the dataspace, given a stochastic model and parameter vector. The network learns the mapping $\theta \rightarrow \log \ell_\Phi(\theta|.)$.

As a side benefit, in line with the methods proposed by *Lueckmann et al., 2019*; *Papamakarios et al., 2019b*, the CNN can in fact act as a surrogate simulator; for the purpose of this paper, we do not exploit this possibility however. Since here we attempt to learn distributions, labels were simple binned empirical likelihoods, and as a loss function we chose the Kullback Leibler (KL) divergence between the network's output distribution and the label distribution (details are given in the 'Materials and methods' section).

## Training specifics

Most of the specifics regarding training procedures are discussed in detail in the 'Materials and methods' section; however, we mention some aspects here to aid readability.

We used 1.5 M and 3 M parameter vectors (based on 100k simulations each) to train the MLP and CNN approach, respectively. We chose these numbers consistently across all models and in fact trained on less examples for the MLP only due to RAM limitations we faced on our machines (which in principle can be circumvented). These numbers are purposely high (but in fact quite achievable with access to a computing cluster, simulations for each model was on the order of hours only) since we were interested in a workable proof of concept. We did not investigate the systematic minimization of training data; however, some crude experiments indicate that a decrease by an order of magnitude did not seriously affect performance.

We emphasize that this is in line with the expressed philosophy of our approach. The point of amortization is to throw a lot of resources at the problem once so that downstream inference is made accessible even on basic setups (a usual laptop). In case simulators are prohibitively expensive even for reasonably sized computer clusters, minimizing training data may gain more relevance. In such scenarios, training the networks will be very cheap compared to simulation time, which implies that retraining with progressively more simulations until one observes asymptotic test performance is a viable strategy.

## Test beds

We chose variations of SSMs common in the cognitive neurosciences as our test bed (*Figure 3*). The range of models we consider permits great flexibility in allowable data distributions (choices and response times). We believe that initial applications are most promising for such SSMs because (1) analytical likelihoods are available for the most common variants (and thus provide an upper-bound benchmark for parameter recovery) and (2) there exist many other interesting variants for which no analytic solution exists.

We note that there is an intermediate case in which numerical methods can be applied to obtain likelihoods for a broader class of models (e.g., *Shinn et al., 2020*). These methods are nevertheless computationally expensive and do not necessarily afford rapid posterior inference. Therefore, amortization via LANs is attractive even for these models. *Figure 3* further outlines this distinction.

We emphasize that our methods are quite general, and any model that generates discrete choices and response times from which simulations can be drawn within a reasonable amount of time can be suitable to the amortization techniques discussed in this paper (given that the model has parameter vectors of dimension roughly $lt_{15}$). In fact, LANs are not restricted to models of reaction time and choice to begin with, even though we focus on these as test beds.

As a general principle, all models tested below are based on stochastic differential equations (SDEs) of the following form:

$$d\mathbf{X}_t = a(t,x)\,dt + b(t,x)\,d\mathbf{B}_t, \quad \mathbf{X}_0 = w$$

where we are concerned with the probabilistic behavior of the particle (or vector of particles) $\mathbf{X}$. The behavior of this particle is driven by $a(t,x)$, an underlying drift function, $b(t,x)$, an underlying noise transformation function, $B_t$, an incremental noise process, and $X_0 = w$, a starting point.

Of interest to us are specifically the properties of the first-passage-time-distributions (FPTD) for such processes, which are needed to compute the likelihood of a given response time/choice pair $\{rt, c\}$. In these models, the exit region of the particle (i.e., the specific boundary it reaches) determines the choice, and the time point of that exit determines the response time. The joint distribution of choices and reaction times is referred to as a FPTD.

Given some exit region $\mathcal{E}$, such FPTDs are formally defined as

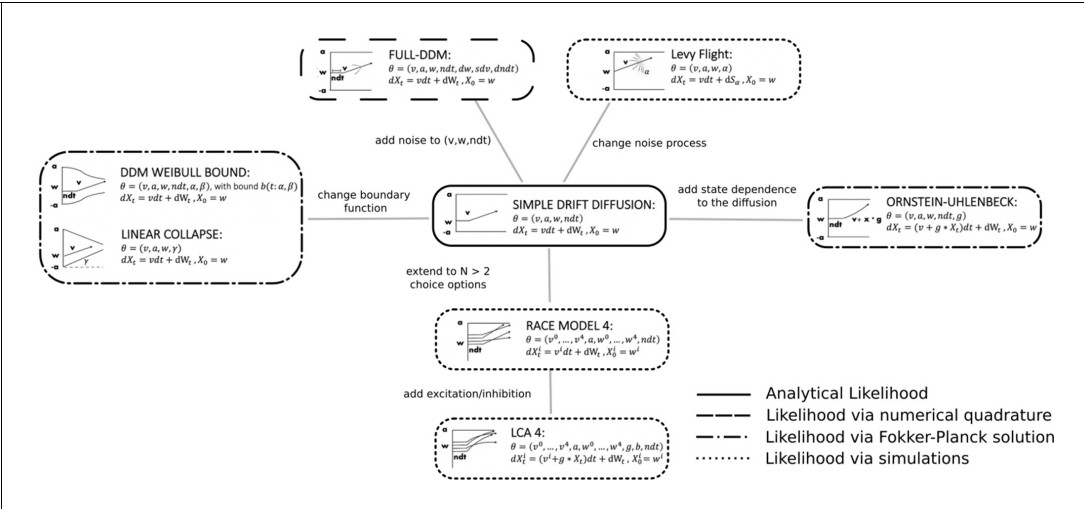

**Figure 3.** Pictorial representation of the stochastic simulators that form our test bed. Our point of departure is the standard simple drift diffusion model (DDM) due to its analytical tractability and its prevalence as the most common sequential sampling model (SSM) in cognitive neuroscience. By systematically varying different facets of the DDM, we test our likelihood approximation networks (LANs) across a range of SSMs for parameter recovery, goodness of fit (posterior predictive checks), and inference runtime. We divide the resulting models into four classes as indicated by the legend. We consider the simple DDM in the *analytical likelihood* (solid line) category, although, strictly speaking, the likelihood involves an infinite sum and thus demands an approximation algorithm introduced by Navarro and Fuss, but this algorithm is sufficiently fast to evaluate so that it is not a computational bottleneck. The full-DDM needs *numerical quadrature* (dashed line) to integrate over variability parameters, which inflates the evaluation time by 1–2 orders of magnitude compared to the simple DDM. Similarly, likelihood approximations have been derived for a range of models using the Fokker–Planck equations (dotted-dashed line), which again incurs nonsignificant evaluation cost. Finally, for some models no approximations exist and we need to resort to computationally expensive *simulations* for likelihood estimates (dotted line). Amortizing computations with LANs can substantially speed up inference for all but the *analytical likelihood* category (but see runtime for how it can even provide speedup in that case for large datasets).

$$f_{\mathcal{E}}(t) = p(\inf_{\tau}\{(X_\tau \in \mathcal{E})\} = t)$$

In words, a first passage time, for a given exit region, is defined as the first time point of entry into the exit region, and the FPTD is the probability, respectively, of such an exit happening at any specified time $t$ (**Ratcliff, 1978**; **Feller, 1968**). Partitioning the exit region into subsets $\mathcal{E}_1, \ldots, \mathcal{E}_n$ (e.g., representing $n$ choices), we can now define the set of defective distributions

$$\{f_{\mathcal{E}_1}(t;\theta), \ldots, f_{\mathcal{E}_N}(t;\theta)\}$$

where $\theta \in \Theta$ describes the collection of parameters driving the process. For every $\mathcal{E}_i$,

$$\int_{[0,+\infty]} f_{\mathcal{E}_i}(t;\theta)dt = P(\mathbf{X} \text{ exits into } \mathcal{E}_i) = P(i \text{ gets chosen})$$

$\{f_{\mathcal{E}_\rangle}, \ldots, f_{\mathcal{E}_\backslash}\}$ jointly define the FPTD such that

$$\sum_{i=1}^{n} \int_{[0,+\infty]} f_{\mathcal{E}_i}(t;\theta)dt = 1$$

These functions $f_{\mathcal{E}_i}$, jointly serve as the likelihood function s.t.

$$\ell(\theta; \{rt, c\}) = f_{\mathcal{E}_c}(t;\theta)$$

For illustration, we focus the general model formulation above to the standard DDM. Details regarding the other models in our test bed are relegated to the 'Materials and methods' section.

To obtain the DDM from the general equation above, we set $a(t,x) = v$ (a fixed drift across time), $b(t,x) = 1$ (a fixed noise variance across time), and $\Delta\mathbf{B} \sim \mathcal{N}(0, \Delta t)$. The DDM applies to the two alternative decision cases, where decision corresponds to the particle crossings of an upper or lower fixed boundary. Hence, $\mathcal{E}_1 = \{\mathbb{R} \geq a\}$ and $\mathcal{E}_2 = \{\mathbb{R} \leq -a\}$, where $a$ is a parameter of the model. The

DDM also includes a normalized starting point $w$ (capturing potential response biases or priors), and finally a nondecision time $\tau$ (capturing the time for perceptual encoding and motor output). Hence, the parameter vector for the DDM is then $\theta = (v, a, w, \tau)$. The SDE is defined as

$$d\mathbf{X}_{\tau+t} = v\,dt + d\mathbf{W}, \ \mathbf{X}_\tau = w$$

The DDM serves principally as a basic proof of concept for us, in that it is a model for which we can compute the exact likelihoods analytically (*Feller, 1968*; *Navarro and Fuss, 2009*).

The other models chosen for our test bed systematically relax some of the fixed assumptions of the basic DDM, as illustrated in *Figure 3*.

We note that many new models can be constructed from the components tested here. As an example of this modularity, we introduce inhibition/excitation to the race model, which gives us the leaky competing accumulator (LCA) (*Usher and McClelland, 2001*). We could then further extend this model by introducing parameterized bounds. We could introduce reinforcement learning parameters to a DDM (*Pedersen and Frank, 2020*) or in combination with any of the other decision models. Again we emphasize that while these diffusion-based models provide a large test bed for our proposed methods, applications are in no way restricted to this class of models.

## Results

### Networks learn likelihood function manifolds

Across epochs of training, both training and validation loss decrease rapidly and remain low (*Figure 4A*) suggesting that overfitting is not an issue, which is sensible in this context. The low validation loss further shows that the network can interpolate likelihoods to specific parameter values it has not been exposed to (with the caveat that it has to be exposed to the same range; no claims are made about extrapolation).

Indeed, a simple interrogation of the learned likelihood manifolds shows that they smoothly vary in an interpretable fashion with respect to changes in generative model parameters (*Figure 4B*). Moreover, *Figure 4C* shows that the MLP likelihoods mirror those obtained by KDEs using 100,000 simulations, even though the model parameter vectors were drawn randomly and thus not trained per se. We also note that the MLP likelihoods appropriately filter out simulation noise (random fluctuations in the KDE empirical likelihoods across separate simulation runs of $100K$ samples each). This observation can also be gleaned from *Figure 4C*, which shows the learned likelihood to sit right at the center of sampled KDEs (note that for each subplot 100 such KDEs were used). As illustrated in the Appendix, these observations hold across all tested models. One perspective on this is to consider the MLP likelihoods as equivalent to KDE likelihoods derived from a much larger number of underlying samples and interpolated. The results for the CNN (not shown to avoid redundancy) mirror the MLP results. Finally, while *Figure 4* depicts the learned likelihood for the simple DDM for illustration purposes, the same conclusions apply to the learned manifolds for all of the tested models (as shown in *Appendix 1—figures 1–6*). Indeed inspection of those manifolds is insightful for facilitating interpretation of the dynamics of the underlying models, how they differ from each other, and the corresponding RT distributions that can be captured.

### Parameter recovery

#### Benchmark: analytical likelihood available

While the above inspection of the learned manifolds is promising, a true test of the method is to determine whether one can perform proper inference of generative model parameters using the MLP and CNN. Such parameter recovery exercises are typically performed to determine whether a given model is identifiable for a given experimental setting (e.g., number of trials, conditions, etc.). Indeed, when parameters are collinear, recovery can be imperfect even if the estimation method itself is flawed (*Wilson and Collins, 2019*; *Nilsson et al., 2011*; *Daw, 2011a*). A Bayesian estimation method, however, should properly assign uncertainty to parameter estimates in these circumstances, and hence it is also important to evaluate the posterior variances over model parameters.

Thus as a benchmark, we first consider the basic DDM for which an arbitrarily close approximation to the analytical likelihood is available (*Navarro and Fuss, 2009*). This benchmark allows us to compare parameter recovery given (1) the analytical likelihood, (2) an approximation to the likelihood

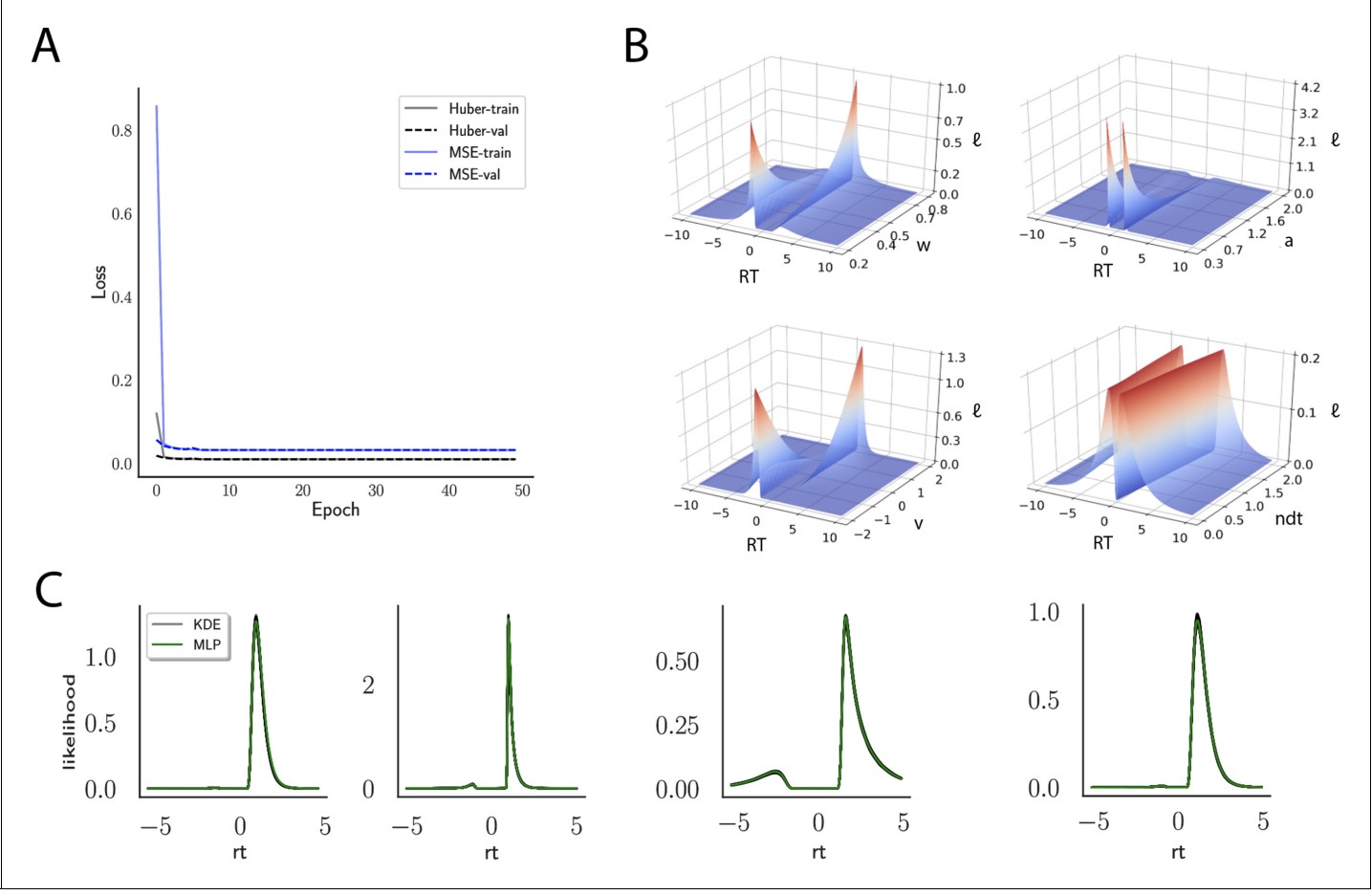

**Figure 4.** Likelihoods and manifolds: DDM. (**A**) shows the training and validation loss for the multilayered perceptron (MLP) for the drift diffusion model across epochs. Training was driven by the Huber loss. The MLP learned the mapping $\{\theta, rt, c\} \mapsto log\ell(\theta|rt, c)$, that is, the log-likelihood of a single-choice/ RT datapoint given the parameters. Training error declines rapidly, and validation loss trailed training loss without further detriment (no overfitting). Please see *Figure 2* and the 'Materials and methods' section for more details about training procedures. (**B**) illustrates the marginal likelihood manifolds for choices and RTs by varying one parameter in the trained region. Reaction times are mirrored for choice options −1, and 1, respectively, to aid visualization. (**C**) shows MLP likelihoods in green for four random parameter vectors, overlaid on top of a sample of 100 kernel density estimate (KDE)-based empirical likelihoods derived from 100k samples each. The MLP mirrors the KDE likelihoods despite not having been explicitly trained on these parameters. Moreover, the MLP likelihood sits firmly at the mean of sample of 100 KDEs. Negative and positive reaction times are to be interpreted as for (**B**).

specified by training an MLP on the analytical likelihood (thus evaluating the potential loss of information incurred by the MLP itself), (3) an approximation to the likelihood specified by training an MLP on KDE-based empirical likelihoods (thus evaluating any further loss incurred by the KDE reconstruction of likelihoods), and (4) an approximate likelihood resulting from training the CNN architecture, on empirical histograms. *Figure 5* shows the results for the DDM.

For the simple DDM and analytical likelihood, parameters are nearly perfectly recovered given $N = 1024$ datapoints ('trials') (*Figure 5A*). Notably, these results are mirrored when recovery is performed using the MLP trained on the analytical likelihood (*Figure 5B*). This finding corroborates, as visually suggested by the learned likelihood manifolds, the conclusion that globally the likelihood function was well behaved. Moreover, only slight reductions in recoverability were incurred when the MLP was trained on the KDE likelihood estimates (*Figure 5C*), likely due to the known small biases in KDE itself (*Turner et al., 2015*). Similar performance is achieved using the CNN instead of MLP (*Figure 5D*).

As noted above, an advantage of Bayesian estimation is that we obtain an estimate of the posterior uncertainty in estimated parameters. Thus, a more stringent requirement is to additionally

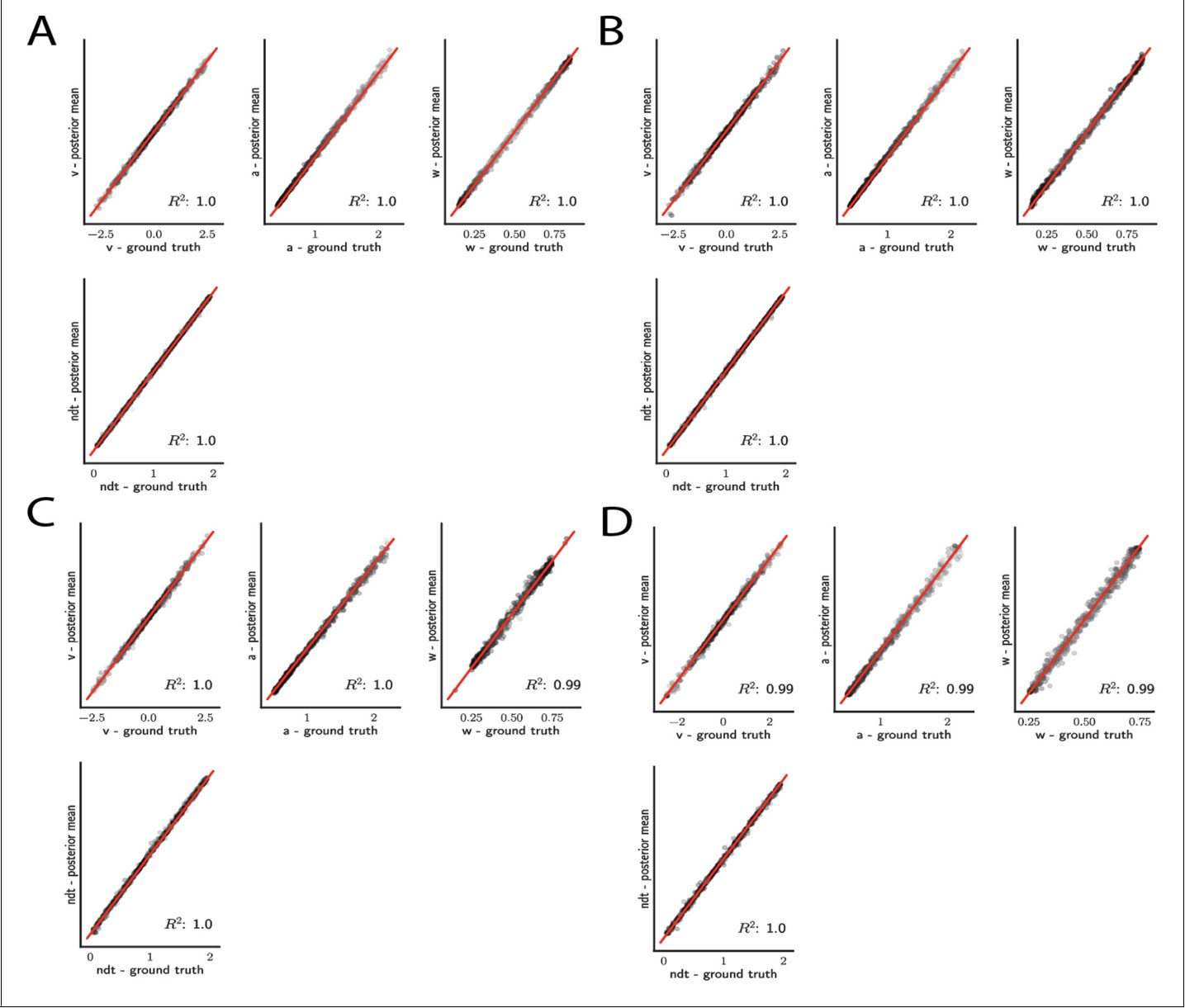

**Figure 5.** Simple drift diffusion model parameter recovery results for (A) analytical likelihood (ground truth), (B) multilayered perceptron (MLP) trained on analytical likelihood, (C) MLP trained on kernel density estimate (KDE)-based likelihoods (100K simulations per KDE), and (D) convolutional neural network trained on binned likelihoods. The results represent posterior means, based on inference over datasets of size $N_1 = 1024$'trials'. Dot shading is based on parameter-wise normalized posterior variance, with lighter shades indicating larger posterior uncertainty of the parameter estimate.

recover the correct posterior variance for a given dataset $\mathcal{D}$ and model $\mathcal{M}$. One can already see visually in *Figure 5C, D* that posterior uncertainty is larger when the mean is further from the ground truth (lighter shades of gray indicate higher posterior variance). However, to be more rigorous one can assess whether the posterior variance is precisely what it should be.

The availability of an analytical likelihood for the DDM, together with our use of sampling methods (as opposed to variational methods that can severely bias posterior variance), allows us to obtain the 'ground truth' uncertainty in parameter estimates. *Figure 6* shows that the sampling from a MLP trained on analytical likelihoods, an MLP trained on KDE-based likelihoods, and a CNN all yield excellent recovery of the variance. For an additional run that involved datasets of size $n = 4096$ instead of $n = 1024$, we observed a consistent decrease in posterior variance across all methods (not shown) as expected.

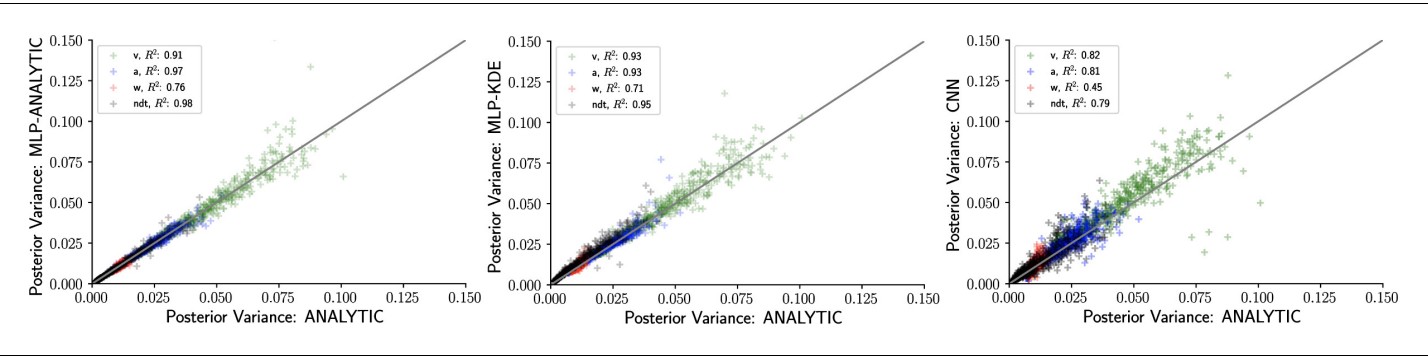

**Figure 6.** Inference using likelihood approximation networks (LANs) recovers posterior uncertainty. Here, we leverage the analytic solution for the drift diffusion model to plot the 'ground truth' posterior variance on the x-axis, against the posterior variance from the LANs on the y-axis. (Left) Multilayered perceptrons (MLPs) trained on the analytical likelihood. (Middle) MLPs trained on kernel density estimate-based empirical likelihoods. (Right) Convolutional neural networks trained on binned empirical likelihoods. Datasets were equivalent across methods for each model (left to right) and involved $n = 1024$ samples.

## No analytical likelihood available

As a proof of concept for the more general ABC setting, we show parameter recovery results for two nonstandard models, the linear collapse (LC) and Weibull models, as described in the 'Test bed' section. The results are summarized in *Figures 7* and *8* and described in more detail in the following two paragraphs.

### Parameter recovery

*Figure 7* shows that both the MLP and CNN methods consistently yield very good to excellent parameter recovery performance for the LC model, with parameter-wise regression coefficients globally above $R^2 > 0.9$. As shown in *Figure 8*, parameter recovery for the Weibull model is less successful, however, particularly for the Weibull collapsing bound parameters. The drift parameter $v$, the starting point bias $w$, and the nondecision time are estimated well; however, the boundary parameters $a$, $\alpha$, and $\beta$ are less well recovered by the posterior mean. Judging by the parameter recovery plot, the MLP seems to perform slightly less well on the boundary parameters when compared to the CNN.

To interrogate the source of the poor recovery of $\alpha$ and $\beta$ parameters, we considered the possibility that the model itself may have issues with identifiability, rather than poor fit. *Figure 8* shows that indeed, for two representative datasets in which these parameters are poorly recovered, the model nearly perfectly reproduces the ground truth data in the posterior predictive RT distributions. Moreover, we find that whereas the individual Weibull parameters are poorly recovered, the net boundary $B(t)$ is very well recovered, particularly when evaluated within the range of the observed dataset. This result is reminiscent of the literature on sloppy models (*Gutenkunst et al., 2007*), where sloppiness implies that various parameter configurations can have the same impact on the data. Moreover, two further conclusions can be drawn from this analysis. First, when fitting the Weibull model, researchers should interpret the bound trajectory as a latent parameter rather than the individual $\alpha$ and $\beta$ parameters per se. Second, the Weibull model may be considered as viable only if the estimated bound trajectory varies sufficiently within the range of the empirical RT distributions. If the bound is instead flat or linearly declining in that range, the simple DDM or LC models may be preferred, and their simpler form would imply that they would be selected by any reasonable model comparison metric. Lastly, given our results the Weibull model could likely benefit from reparameterization if the desire is to recover individual parameters rather than the bound trajectory $B(t)$. Given the common use of this model in collapsing bound studies (*Hawkins et al., 2015*) and that the bound trajectories are nevertheless interpretable, we leave this issue for future work.

The Appendix shows parameter recovery studies on a number of other stochastic simulators with non-analytical likelihoods, described in the 'Test bed' section. The appendices show tables of parameter-wise recovery $R^2$ for all models tested. In general, recovery ranges from good to excellent. Given the Weibull results above, we attribute the less good recovery for some of these models to

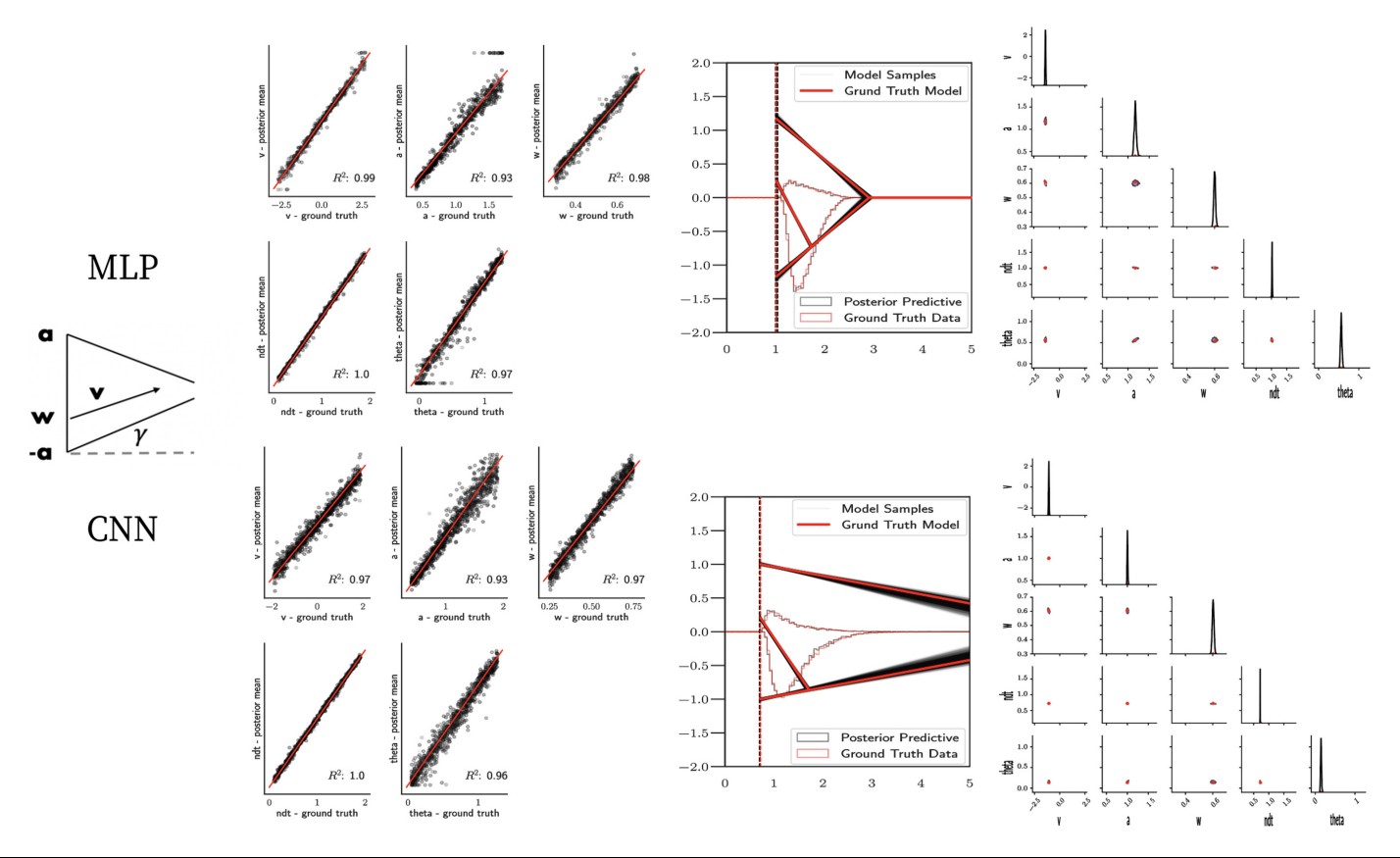

**Figure 7.** Linear collapse model parameter recovery and posterior predictives. (Left) Parameter recovery results for the multilayered perceptron (top) and convolutional neural network (bottom). (Right) Posterior predictive plots for two representative datasets. Model samples of all parameters (black) match those from the true generative model (red), but one can see that for the lower dataset, the bound trajectory is somewhat more uncertain (more dispersion of the bound). In both cases, the posterior predictive (black histograms) is shown as predicted choice proportions and RT distributions for upper and lower boundary responses, overlaid on top of the ground truth data (red; hardly visible since overlapping/matching).

identifiability issues and specific dataset properties rather than to the method per se. We note that our parameter recovery studies here are in general constrained to the simplest inference setting equivalent to a single-subject, single-condition experimental design. Moreover, we use uninformative priors for all parameters of all models. Thus, these results provide a lower bound on parameter recoverability, provided of course that the datasets were generated from the valid parameter ranges on which the networks were trained; see Section 0.10 for how recovery can benefit from more complex experimental designs with additional task conditions, which more faithfully represents the typical inference scenario deployed by cognitive neuroscientists. Lastly, some general remarks about the parameter recovery performance. A few factors can negatively impact how well one can recover parameters. First, if the model generally suffers from identifiability issues, the resulting tradeoffs in the parameters can lead the MCMC chain to get stuck on the boundary for one or more parameters. This issue is endemic to all models and unrelated to likelihood-free methods or LANs, and should at best be attacked at the level of reparameterization (or a different experimental design that can disentangle model parameters). Second, if the generative parameters of a dataset are too close to (or beyond) the bounds of the trained parameter space, we may also end with a chain that gets stuck on the boundary of the parameter space. We confronted this problem by training on parameter spaces that yield response time distributions that are broader than typically observed experimentally for models of this class, while also excluding obviously defective parameter setups. Defective parameter setups were defined in the context of our applications as parameter vectors that generate data that never allow one or the other choice to occur (as in $p(c) << \frac{1}{100,000}$, data that concentrates more

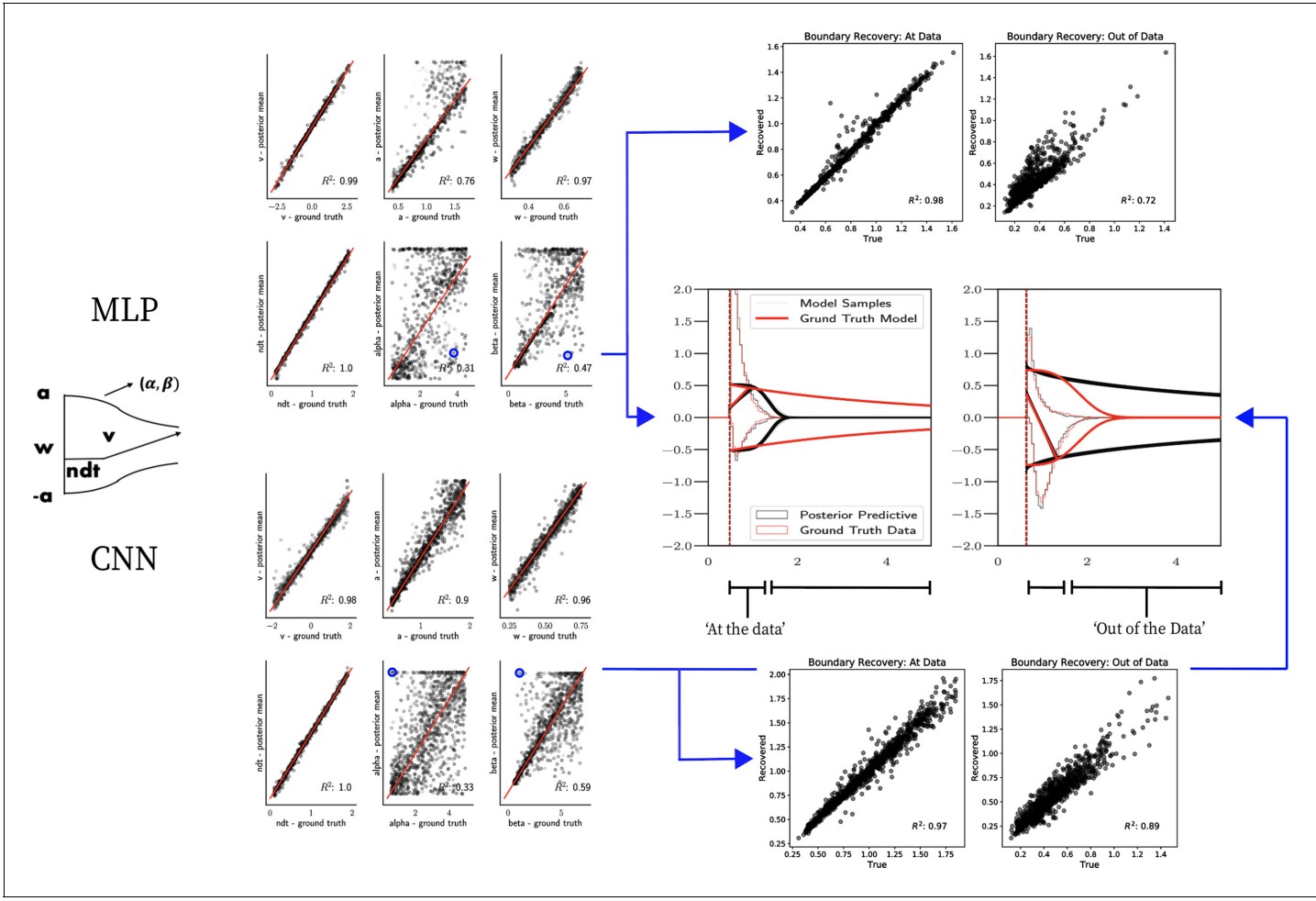

**Figure 8.** Weibull model parameter recovery and posterior predictives. (Left) Parameter recovery results for the multilayered perceptron (top) and convolutional neural network (bottom). (Right) Posterior predictive plots for two representative datasets in which parameters were poorly estimated (denoted in blue on the left). In these examples, model samples (black) recapitulate the generative parameters (red) for the nonboundary parameters, and the recovered bound trajectory is poorly estimated relative to the ground truth, despite excellent posterior predictives in both cases (RT distributions for upper and lower boundary, same scheme as *Figure 7*). Nevertheless, one can see that the net decision boundary is adequately recovered within the range of the RT data that are observed. Across all datasets, the net boundary $B(t) = a * \exp\left(-\frac{t^\alpha}{\beta}\right)$ is well recovered within the range of the data observed, and somewhat less so outside of the data, despite poor recovery of individual Weibull parameters α and β.

than half of the reaction times within a single 1 ms bin and data that generated mean reaction times beyond 10 s). These guidelines were chosen as a mix of basic rationale and domain knowledge regarding usual applications of DDMs to experimental data. As such, the definition of defective data may depend on the model under consideration.

## Runtime

A major motivation for this work is the amortization of network training time during inference, affording researchers the ability to test a variety of theoretically interesting models for linking brain and behavior without large computational cost. To quantify this advantage, we provide some results on the posterior sampling runtimes using (1) the MLP with slice sampling (*Neal, 2003*) and (2) CNN with iterated importance sampling.

The MLP timings are based on slice sampling (*Neal, 2003*), with a minimum of $n = 2000$ samples. The sampler was stopped at some $n >= 2000$, for which the Geweke statistic (*Geweke, 1992*) indicated convergence (the statistic was computed once every 100 samples for $n >= 2000$). Using an alternative sampler, based on differential evolution Markov chain Monte Carlo (DEMCMC) and

stopped when the Gelman–Rubin $\hat{R}<1.1$ (*Gelman and Rubin, 1992*), yielded very similar timing results and was omitted in our figures.

For the reported importance sampling runs, we used 200K importance samples per iteration, starting with $\gamma$ values of 64, which was first reduced to 1 where in iteration $i$, $\gamma_i = \frac{64}{2^{i-1}}$, before a stopping criterion based on relative improvement of the confusion metric was used.

*Figure 9A* shows that all models can be estimated in the order of hundreds of seconds (minutes), comprising a speed improvement of at least two orders of magnitude compared to traditional ABC methods using KDE during inference (i.e., the PDA method motivating this work; *Turner et al., 2015*). Indeed, this estimate is a lower bound on the speed improvement: we extrapolate only the observed difference between network evaluation and online simulations, ignoring the additional cost of constructing and evaluating the KDE-based likelihood. We decided to use this benchmark because it provides a fairer comparison to more recent PDA approaches in which the KDE evaluations can be sped up considerably (*Holmes, 2015*).

Notably, due to its potential for parallelization (especially on GPUs), our NN methods can even induce performance speedups relative to analytical likelihood evaluations. Indeed, *Figure 9B* shows that as the dataset grows runtime is significantly faster than even a highly optimized cython implementation of the Navarro Fuss algorithm (*Navarro and Fuss, 2009*) for evaluation of the analytic DDM likelihood. This is also noteworthy in light of the full-DDM (as described in the 'Test bed' section), for which it is currently common to compute the likelihood term via quadrature methods, in turn based on repeated evaluations of the Navarro Fuss algorithm. This can easily inflate the

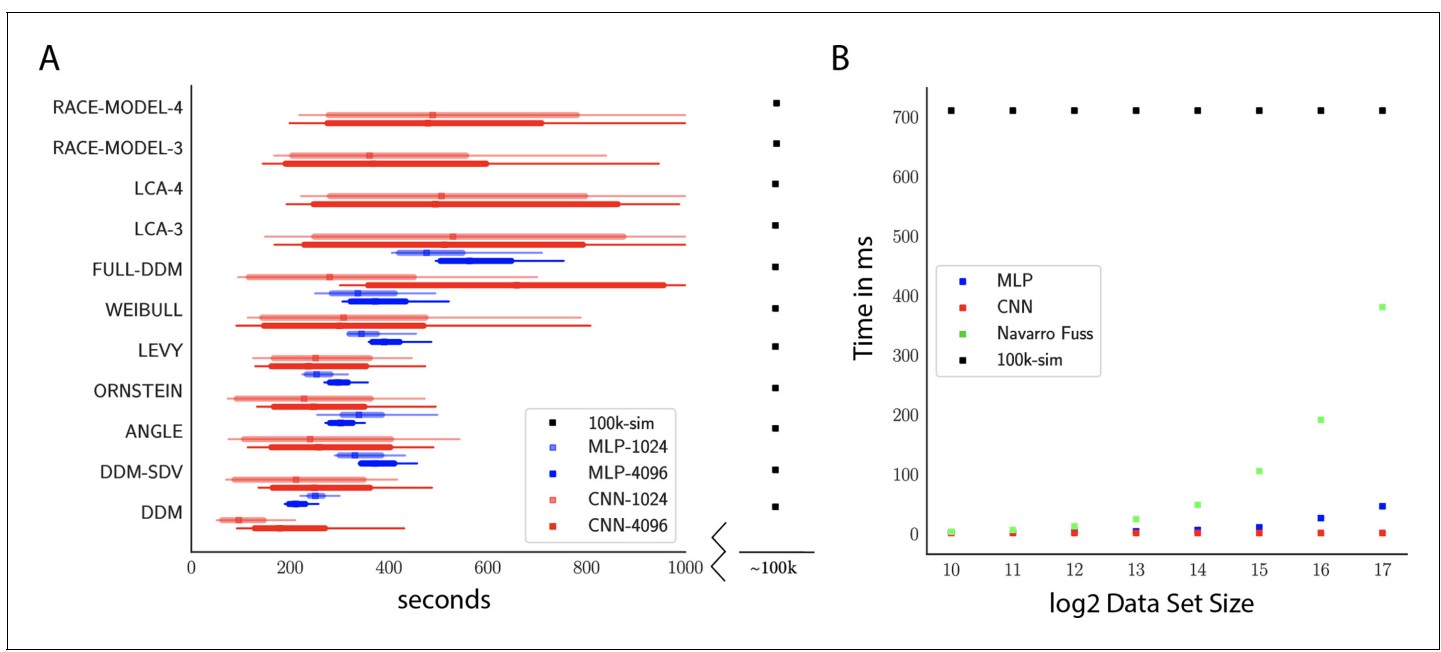

**Figure 9.** Computation times. (**A**) Comparison of sampler timings for the multilayered perceptron (MLP) and convolutional neural network (CNN) methods, for datasets of size 1024 and 4096 (respectively MLP-1024, MLP-4096, CNN-1024, CNN-4096). For comparison, we include a lower bound estimate of the sample timings using traditional PDA approach during online inference (using 100k online simulations for each parameter vector). 100K simulations were used because we found this to be required for sufficiently smooth likelihood evaluations and is the number of simulations used to train our networks; fewer samples can of course be used at the cost of worse estimation, and only marginal speedup since the resulting noise in likelihood evaluations tends to prevent chain mixing; see *Holmes, 2015*. We arrive at 100k seconds via simple arithmetic. It took our slice samplers on average approximately 200k likelihood evaluations to arrive at 2000 samples from the posterior. Taking 500 ms * 200,000 gives the reported number. Note that this is a generous but rough estimate since the cost of data simulation varies across simulators (usually quite a bit higher than the drift diffusion model [DDM] simulator). Note further that these timings scale linearly with the number of participants and task conditions for the online method, but not for likelihood approximation networks, where they can be in principle be parallelized. (**B**) compares the timings for obtaining a single likelihood evaluation for a given dataset. MLP and CNN refer to Tensorflow implementations of the corresponding networks. Navarro Fuss refers to a cython (*Behnel et al., 2010*) (cpu) implementation of the algorithm suggested (*Navarro and Fuss, 2009*) for fast evaluation of the analytical likelihood of the DDM. 100k-sim refers to the time it took a highly optimized cython (cpu) version of a DDM sampler to generate 100k simulations (averaged across 100 parameter vectors).

evaluation time by 1–2 orders of magnitude. In contrast, evaluation times for the MLP and CNN are only marginally slower (as a function of the slightly larger network size in response to higher dimensional inputs). We confirm (omitted as separate figure) from experiments with the HDDM Python toolbox that our methods end up approximately $10 - 50$ times faster for the full-DDM than the current implementation based on numerical integration, maintaining comparable parameter recovery performance. We strongly suspect there to be additional remaining potential for performance optimization.

### Hierarchical inference

One of the principal benefits of LANs is that they can be directly extended – without further training – to arbitrary hierarchical inference scenarios, including those in which (1) individual participant parameters are drawn from group distributions, (2) some parameters are pooled and others separated across task conditions, and (3) neural measures are estimated as regressors on model parameters (*Figure 10*). Hierarchical inference is critical for improving parameter estimation particularly for realistic cognitive neuroscience datasets in which thousands of trials are not available for each participant and/or where one estimates impacts of noisy physiological signals onto model parameters (*Wiecki et al., 2013*; *Boehm et al., 2018*; *Vandekerckhove et al., 2011*; *Ratcliff and Childers, 2015*).

To provide a proof of concept, we developed an extension to the HDDM Python toolbox (*Wiecki et al., 2013*), widely used for hierarchical inference of the DDM applied to such settings. Lifting the restriction of previous versions of HDDM to only DDM variants with analytical likelihoods, we imported the MLP likelihoods for all two-choice models considered in this paper. Note that GPU-based computation is supported out of the box, which can easily be exploited with minimal overhead using free versions of Google's Colab notebooks. We generally observed GPUs to improve speed approximately fivefold over CPU-based setups for the inference scenarios we tested. Preliminary access to this interface and corresponding instructions can be found at https://github.com/lnccbrown/lans/tree/master/hddmnn_tutorial (copy archived at swh:1:rev:e3369b9df138c75d0e490-be0c48c53ded3e3a1d6); *Fengler, 2021*.

*Figure 11* shows example results from hierarchical inference using the LC model, applied to synthetic datasets comprising 5 and 20 subjects (a superset of participants). Recovery of individual parameters was adequate even for five participants, and we also observe the expected improvement of recovery of the group-level parameters μ and σ for 20 participants.

*Figure 12* shows an example that illustrates how parameter recovery is affected when a dataset contains multiple experimental conditions (e.g., different difficulty levels). It is common in such scenarios to allow task conditions to affect a single (or subset)-model parameter (in the cases shown: *v*), while other model parameters are pooled across conditions. As expected, for both the full-DDM (A) and the Levy model (B), the estimation of global parameters is improved when increasing the number of conditions from 1 to 5 to 10 (left to right, where the former are subsets of the latter datasets). These experiments confirm that one can more confidently estimate parameters that are otherwise difficult to estimate such as the noise α in the Levy model and *sv* the standard deviation of the drift in the full-DDM.

Both of these experiments provide evidence that our MLPs provide approximate likelihoods which behave in accordance with what is expected from proper analytical methods, while also demonstrating their robustness to other samplers (i.e., we used HDDM slice samplers without further modification for all models).

We expect that proper setting of prior distributions (uniform in our examples) and further refinements to the slice sampler settings (to help mode discovery) can improve these results even further. We include only the MLP method in this section since it is most immediately amenable to the kind of trial-by-trial-level analysis that HDDM is designed for. We plan to investigate the feasibility of including the CNN method into HDDM in future projects.

## Discussion

Our results demonstrate the promise and potential of amortized LANs for Bayesian parameter estimation of neurocognitive process models. Learned manifolds and parameter recovery experiments

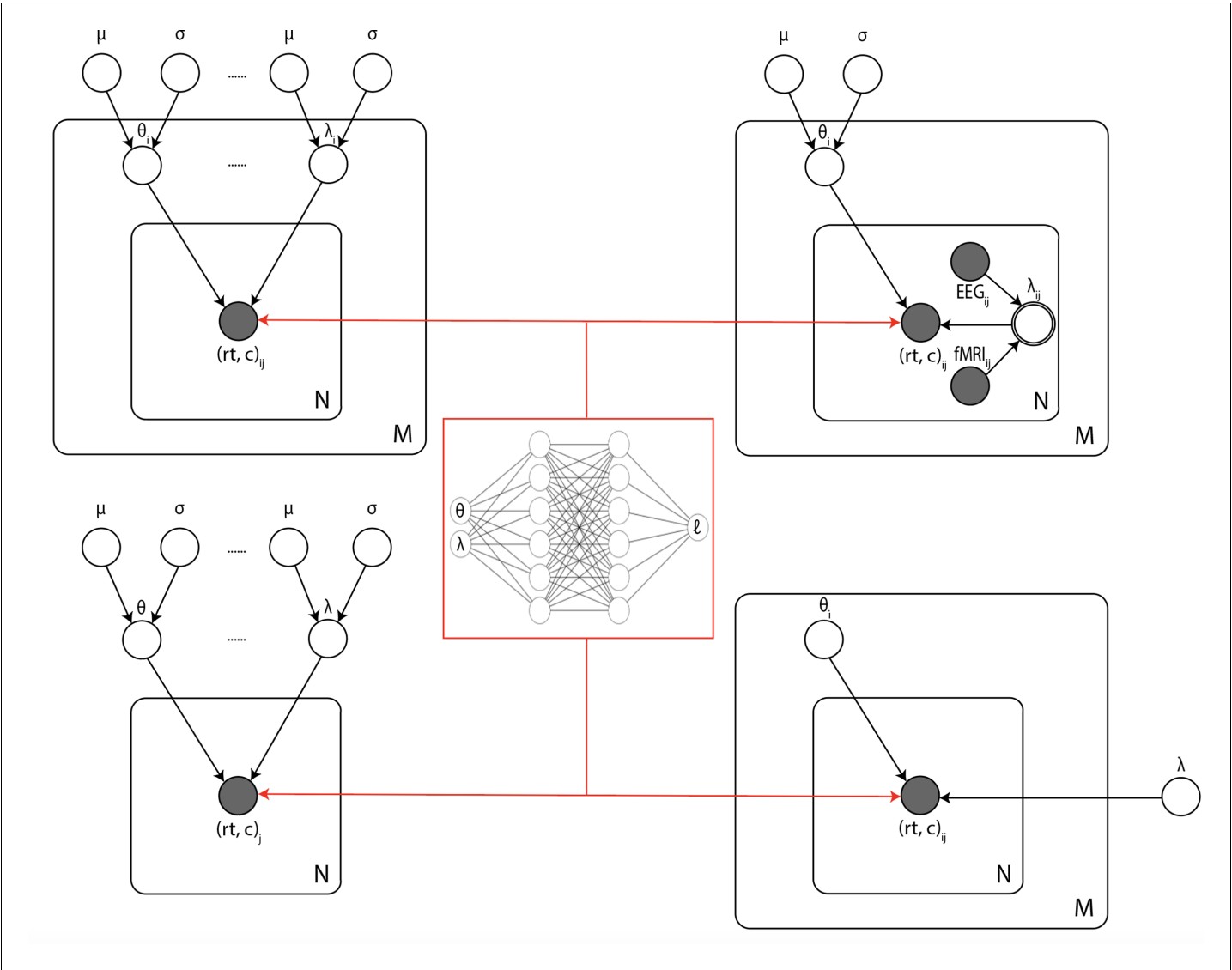

**Figure 10.** Illustration of the common inference scenarios applied in the cognitive neurosciences and enabled by our amortization methods. The figure uses standard plate notation for probabilistic graphical models. White single circles represent random variables, white double circles represent variables computed deterministically from their inputs, and gray circles represent observations. For illustration, we split the parameter vector of our simulator model (which we call θ in the rest of the paper) into two parts θ and λ since some, but not all, parameters may sometimes vary across conditions and/or come from global distribution. (Upper left) Basic hierarchical model across M participants, with N observations (trials) per participant. Parameters for individuals are assumed to be drawn from group distributions. (Upper right) Hierarchical models that further estimate the impact of trial-wise neural regressors onto model parameters. (Lower left) Nonhierarchical, standard model estimating one set of parameters across all trials. (Lower right) Common inference scenario in which a subset of parameters (θ) are estimated to vary across conditions M, while others (λ) are global. Likelihood approximation networks can be immediately repurposed for all of these scenarios (and more) without further training.

showed successful inference using a range of network architectures and posterior sampling algorithms, demonstrating the robustness of the approach.

Although these methods are extendable to any model of similar complexity, we focused here on a class of SSMs, primarily because the most popular of them – the DDM – has an analytic solution, and is often applied to neural and cognitive data. Even slight departures from the standard DDM framework (e.g., dynamic bounds or changes in the noise distribution) are often not considered for full Bayesian inference due to the computational complexity associated with traditional ABC methods. We provide access to the learned likelihood functions (in the form of network weights) and code to enable users to fit a variety of such models with orders of magnitude speedup (minutes

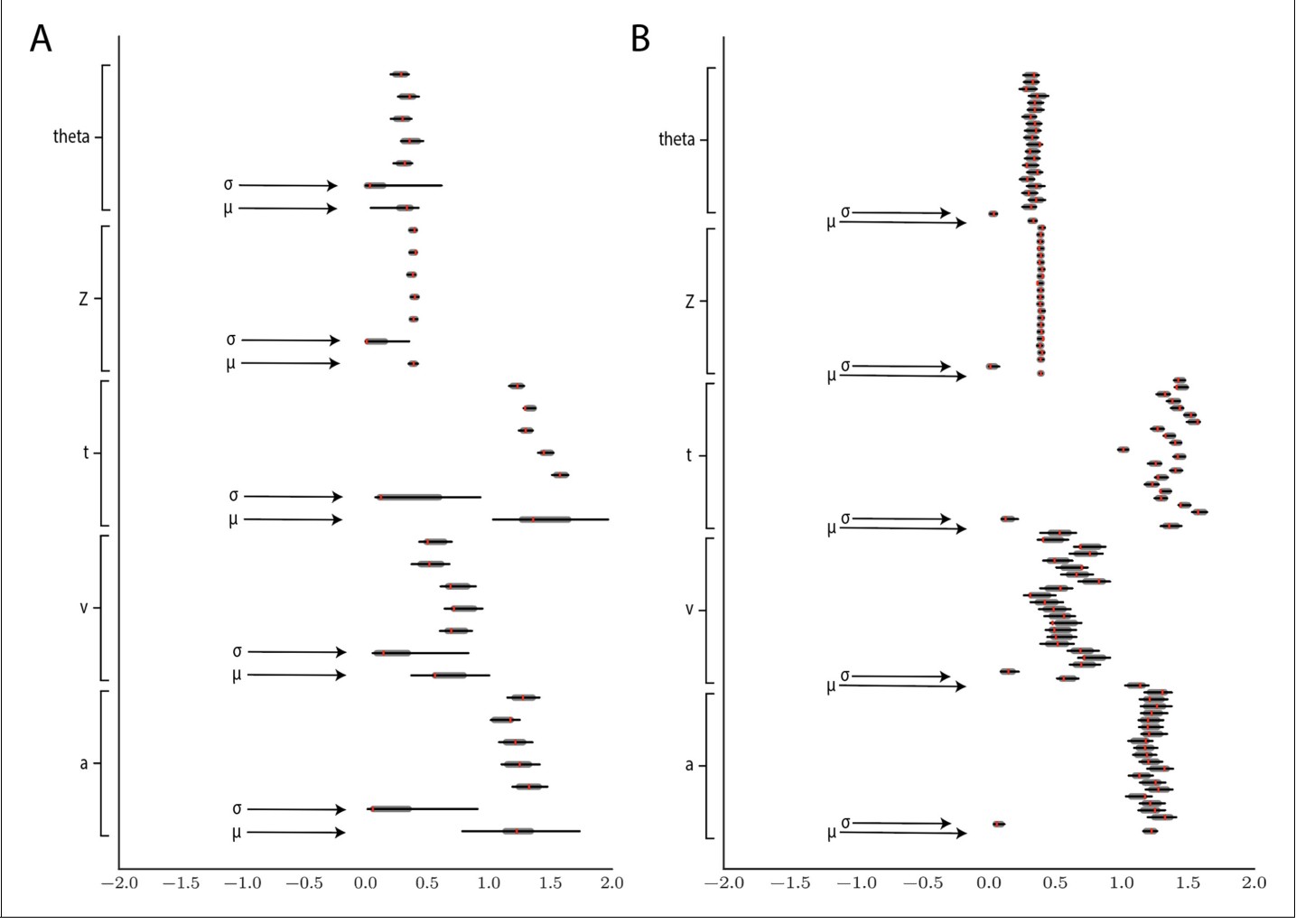

**Figure 11.** Hierarchical inference results using the multilayered perceptron likelihood imported into the HDDM package. (**A**) Posterior inference for the linear collapse model on a synthetic dataset with 5 participants and 500 trials each. Posterior distributions are shown with caterpillar plots (thick lines correspond to $5 - 95$ percentiles, thin lines correspond to $1 - 99$ percentiles) grouped by parameters (ordered from above $\{subject_1, ..., subject_n, \mu_{group}\sigma_{group}\}$). Ground truth simulated values are denoted in red. (**B**) Hierarchical inference for synthetic data comprising 20 participants and 500 trials each. $\mu$ and $\sigma$ indicate the group-level mean and variance parameters. Estimates of group-level posteriors improve with more participants as expected with hierarchical methods. Individual-level parameters are highly accurate for each participant in both scenarios.

instead of days). In particular, we provided an extension to the commonly used HDDM toolbox (*Wiecki et al., 2013*) that allows users to apply these models to their own datasets immediately. We also provide access to code that would allow users to train their own likelihood networks and perform recovery experiments, which can then be made available to the community.

We offered two separate approaches with their own relative advantages and weaknesses. The MLP is suited for evaluating likelihoods of individual observations (choices, response times) given model parameters, and as such can be easily extended to hierarchical inference settings and trial-by-trial regression of neural activity onto model parameters. We showed that importing the MLP likelihood functions into the HDDM toolbox affords fast inference over a variety of models without tractable likelihood functions. Moreover, these experiments demonstrated that use of the NN likelihoods even confers a performance speedup over the analytical likelihood function – particularly for the full-DDM, which otherwise required numerical methods on top of the analytical likelihood function for the simple DDM.

Conversely, the CNN approach is well suited for estimating likelihoods across parameters for entire datasets in parallel, as implemented with importance sampling. More generally and implying

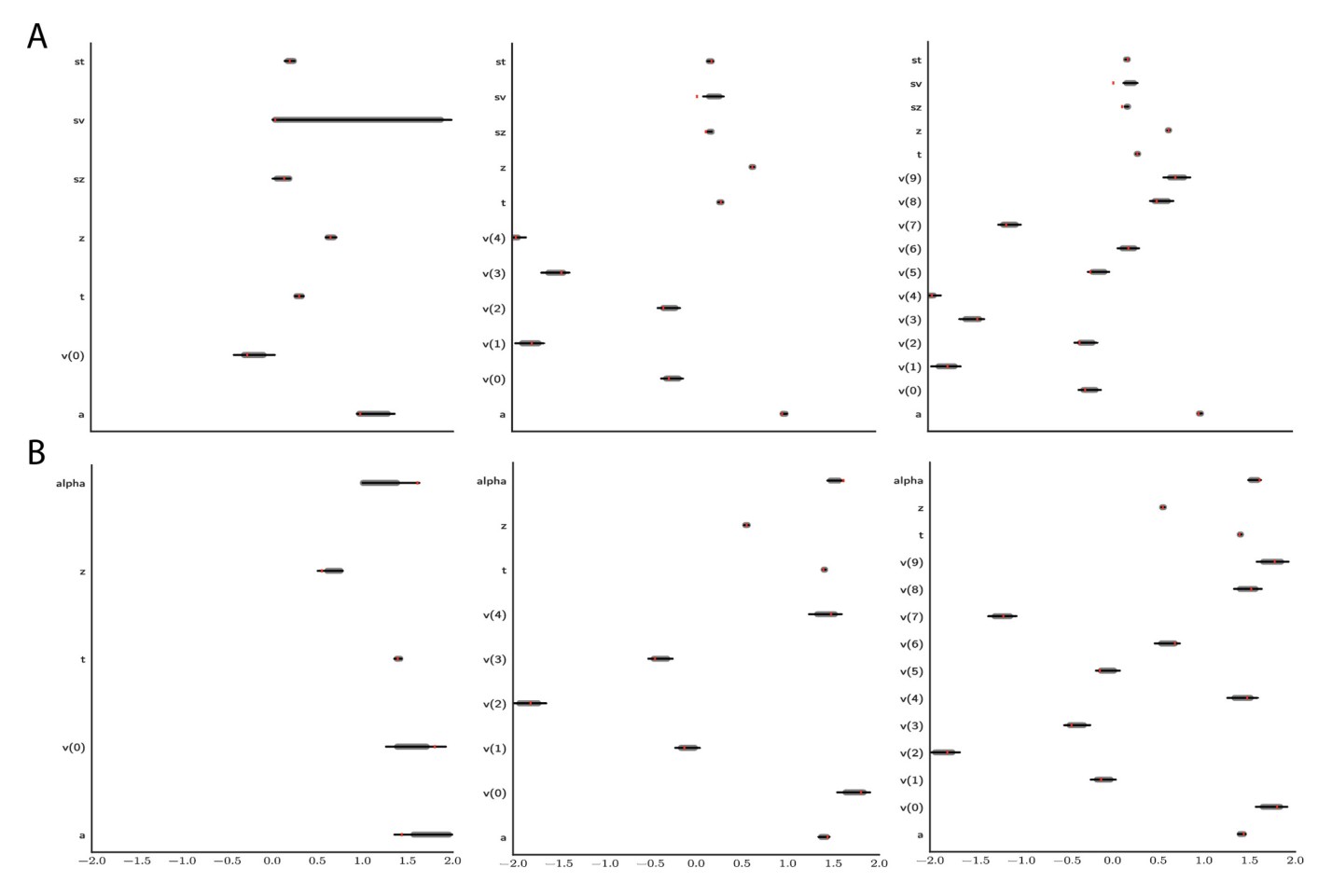

**Figure 12.** Effect of multiple experimental conditions on inference. The panel shows an example of posterior inference for 1, (left), 5 (middle), and 10 (right) conditions. (A) and (B) refer to the full drift diffusion model (DDM) and Levy model, respectively. The drift parameter $v$ is estimated to vary across conditions, while the other parameters are treated as global across conditions. Inference tends to improve for all global parameters when adding experimental conditions. Importantly, this is particularly evident for parameters that are otherwise notoriously difficult to estimate such as $sv$ (trial-by-trial variance in drift in the full-DDM) and $\alpha$ (the noise distribution in the Levy model). Red stripes show the ground truth values of the given parameters.

potential further improvements, any sequential MC method may be applied instead. These methods offer a more robust path to sampling from multimodal posteriors compared to MCMC, at the cost of the curse of dimensionality, rendering them potentially less useful for highly parameterized problems, such as those that require hierarchical inference. Moreover, representing the problem directly as one of learning probability distributions and enforcing the appropriate constraints by design endows the CNN approach with a certain conceptual advantage. Finally, we note that in principle (with further improvements) trial-level inference is possible with the CNN approach, and vice versa, importance sampling can be applied to the MLP approach.

In this work, we employed sampling methods (MCMC and importance sampling) for posterior inference because in the limit they are well known to allow for accurate estimation of posterior distributions on model parameters, including not only mean estimates but their variances and covariances. Accurate estimation of posterior variances is critical for any hypothesis testing scenario because it allows one to be confident about the degree of uncertainty in parameter estimates. Indeed, we showed that for the simple DDM we found that posterior inference using our networks yielded nearly perfect estimation of the variances of model parameters (which are available due to the analytic solution). Of course, our networks can also be deployed for other estimation methods even more rapidly: they can be immediately used for MLE via gradient descent or within other approximate inference methods, such as variational inference (see *Acerbi, 2020* for a related approach).

Other approaches exist for estimating generalized diffusion models. A recent example, not discussed thus far, is the pyDDM Python toolbox (*Shinn et al., 2020*), which allows MLE of generalized drift diffusion models. The underlying solver is based on the Fokker–Planck equations, which allow access to approximate likelihoods (where the degree of approximation is traded off with computation time/discretization granularity) for a flexible class of diffusion-based models, notably allowing arbitrary evidence trajectories, starting point, and nondecision time distributions. However, to incorporate trial-by-trial effects would severely inflate computation time (on the order of the number of trials) since the solver would have to operate on a trial-by-trial level. Moreover, any model that is not driven by Gaussian diffusion, such as the Levy model we considered here or the linear ballistic accumulator, is out of scope with this method. In contrast, LANs can be trained to estimate any such model, limited only by the identifiability of the generative model itself. Finally, pyDDM does not afford full Bayesian estimation and thus quantification of parameter uncertainty and covariance.

We moreover note that our LAN approach can be useful even if the underlying simulation model admits other likelihood approximations, regardless of trial-by-trial effect considerations, since a forward pass through a LAN may be speedier. Indeed, we observed substantial speedups in HDDM for using our LAN method to the full-DDM, for which numerical methods were previously needed to integrate over inter-trial variability.

We emphasize that our test bed application to SSMs does not delimit the scope of application of LANs. Neither are reasonable architectures restricted to MLPs and CNNs (see *Lueckmann et al., 2019*; *Papamakarios et al., 2019b* for related approaches that use completely different architectures). Models with high-dimensional (roughly $gt_{15}$) parameter spaces may present a challenge for our global amortization approach due to the curse of dimensionality. Further, models with discrete parameters of high cardinality may equally present a given network with training difficulties. In such cases, other methods may be preferred over likelihood amortization generally (e.g., *Acerbi, 2020*); given that this is an open and active area of research, we can expect surprising developments that may in fact turn the tide again in the near future.

Despite some constraints, this still leaves a vast array of models in reach for LANs, of which our test bed can be considered only a small beginning.

By focusing on LANs, our approach affords the flexibility of networks serving as plug-ins for hierarchical or arbitrarily complex model extensions. In particular, the networks can be immediately transferred, without further training, to arbitrary inference scenarios in which researchers may be interested in evaluating links between neural measures and model parameters, and to compare various assumptions about whether parameters are pooled and split across experimental manipulations. This flexibility in turn sets our methods apart from other amortization and NN-based ABC approaches offered in the statistics, machine learning, and computational neuroscience literature (*Papamakarios and Murray, 2016*; *Papamakarios et al., 2019a*; *Gonçalves et al., 2020*; *Lueckmann et al., 2019*; *Radev et al., 2020b*), while staying conceptually extremely simple. Instead of focusing on extremely fast inference for very specialized inference scenarios, our approach focuses on achieving speedy inference while not implicitly compromising modeling flexibility through amortization step.

Closest to our approach is the work of *Lueckmann et al., 2019*, and *Papamakarios et al., 2019b*, both of which attempt to target the likelihood with a neural density estimator. While flexible, both approaches imply the usage of summary statistics, instead of a focus on trial-wise likelihood functions. Our work can be considered a simple alternative with explicit focus on trial-wise likelihoods.

Besides deep learning-based approaches, another major machine learning-inspired branch of the ABC literature concerns log-likelihood and posterior approximations via Gaussian process surrogates (GPSs) (*Meeds and Welling, 2014*; *Järvenpää et al., 2018*; *Acerbi, 2020*). A major benefit of GPSs lies in the ability for clever training data selection via active learning since such GPSs allow uncertainty quantification out of the box, which in turn can be utilized for the purpose of targeting high-uncertainty regions in parameter space. GPS-based computations scale with the number of training examples, however, which make them much more suitable for minimizing the computational cost for a given inference scenario than facilitating global amortization as we suggest in this paper (for which one usually need larger sets of training data than can traditionally be handled efficiently by GPS). Again when our approach is applicable, it will offer vastly greater flexibility once a LAN is trained.

## Limitations and future work

There are several limitations of the methods presented in this paper, which we hope to address in future work. While allowing for great flexibility, the MLP approach suffers from the drawback that we do not enforce (or exploit) the constraint that $\hat{\ell}(\theta|\mathbf{x})$ is a valid probability distribution, and hence the networks have to learn this constraint implicitly and approximately. Enforcing this constraint has the potential to improve estimation of tail probabilities (a known issue for KDE approaches to ABC more generally; *Turner et al., 2015*).

The CNN encapsulation exploits the fact that $\int_\mathcal{X} \hat{\ell}(\theta|\mathbf{x})d\mathbf{x} = 1$, however, makes estimation of trial-by-trial effects more resource hungry. We plan to investigate the potential of the CNN for trial-by-trial estimation in future research.

A potential solution that combines the strengths of both the CNN and MLP methods is to utilize mixture density networks to encapsulate the likelihood functions. We are currently exploring this avenue. Mixture density networks have been successfully applied in the context of ABC (*Papamakarios and Murray, 2016*); however, training can be unstable without extra care (*Guillaumes, 2017*). Similarly, invertible flows (*Rezende and Mohamed, 2015*) and/or mixture density networks *Bishop, 1994* may be used to learn likelihood functions (*Papamakarios et al., 2019b*; *Lueckmann et al., 2019*); however, the philosophy remains focused on distributions of summary statistics for single datasets. While impressive improvements have materialized at the intersection of ABC and deep learning methods (*Papamakarios et al., 2019a*; *Greenberg et al., 2019*; *Gonçalves et al., 2020*) (showing some success with posterior amortization for models up to 30 parameters, but restricted to a local region of high posterior density in the resulting parameter space), generally less attention has been paid to amortization methods that are not only of case-specific efficiency but sufficiently modular to serve a large variety of inference scenarios (e.g., *Figure 10*). This is an important gap that we believe the popularization of the powerful ABC framework in the domain of experimental science hinges upon. A second and short-term avenue for future work is the incorporation of our presented methods into the HDDM Python toolbox (*Wiecki et al., 2013*) to extend its capabilities to a larger variety of SSMs. Initial work in this direction has been completed, the alpha version of the extension being available in the form of a tutorial at https://github.com/lnccbrown/lans/tree/master/hddmnn_tutorial.

Our current training pipeline can be further optimized on two fronts. First, no attempt was made to minimize the size of the network needed to reliably approximate likelihood functions so as to further improve computational speed. Second, little attempt was made to optimize the amount of training provided to networks. For the models explored here, we found it sufficient to simply train the networks for a very large number of simulated datapoints such that interpolation across the manifold was possible. However, as model complexity increases, it would be useful to obtain a measure of the networks' uncertainty over likelihood estimates for any given parameter vector. Such uncertainty estimates would be beneficial for multiple reasons. One such benefit would be to provide a handle on the reliability of sampling, given the parameter region. Moreover, such uncertainty estimates could be used to guide the online generation of training data to train the networks in regions with high uncertainty. At the intersection of ABC and NNs, active learning has been explored via uncertainty estimates based on network ensembles (*Lueckmann et al., 2019*). We plan to additionally explore the use of Bayesian NNs, which provide uncertainty over their weights, for this purpose (*Neal, 1995*).

One more general shortcoming of our methods is the reliance on empirical likelihoods for training, which in turn are based on a fixed number of samples across parameter vectors, just as the PDA method proposed by *Turner et al., 2015*. Recently, this approach has been criticized fundamentally on grounds of producing bias in the generated KDE-based likelihood estimates (*van Opheusden et al., 2020*). A reduction of the approximate likelihood problem to one of inverse binomial sampling was proposed (*van Opheusden et al., 2020*), which will generate unbiased likelihood estimates. To address this concern, we will investigate adaptive strategies for the selection of the simulations count $n$. We however highlight two points here that aim to put the promise of unbiased likelihoods in perspective. First, our networks add interpolation to the actual estimation of a likelihood. Likelihoods close in parameter space therefore share information that translates into an effectively higher simulation count than the 100k chosen to construct each empirical likelihood used for training. Quantifying this benefit precisely we leave for future research; however, we suspect, as

suggested by *Figure 4*, that it may be substantial. Second, while we generally acknowledge that bias in the tails remains somewhat of an issue in our approach, resolution is at best partial even in the proposed methods of *van Opheusden et al., 2020*. For the estimation of parameters for which a given datapoint is extremely unlikely (i.e., the data is generally unlikely under the model), the authors suggest to threshold the simulation count so that their algorithm is guaranteed to stop. This effectively amounts to explicitly allowing for bias again. As another alternative, the authors suggest to introduce a lapse rate in the generative model, which the LAN approach can accommodate as well. However, the introduction of a lapse rate does not deal with tail events directly either, but rather assumes that tail events are unrelated to the process of interest. This in turn will render a lower but fixed number of simulations $N$ feasible for training LANs as well. This is notwithstanding the desirable minimization of simulation times even for high likelihood events, especially when trial-wise simulations are in fact necessary (which tends to be in cases where amortization with LANs is a priori not a good computational strategy to begin with). Hence, although the inverse binomial sampling approach is elegant conceptually, excessive computation remains an issue when we need accurate estimates of the probability of actual tail events. Generally, however, we maintain it is desirable and important for future work to make use of the otherwise great potential of adaptive sampling to minimize total computational cost.

Furthermore, we relegate to future research proper exploitation of the fact that LANs are by design differentiable in the parameters. We are currently working on an integration of LANs with Tensorflow probability (*Abadi et al., 2016*), utilizing autograd to switch our MCMC method to the gradient-based NUTS sampler (*Hoffman and Gelman, 2014*). The main benefits of this sampler are robust mixing behavior, tolerance for high levels of correlations in the parameter space, while at the same time maintaining the ability to sample from high-dimensional posteriors. High level of correlations in posteriors is traditionally an Achilles' heel of the otherwise robust coordinate-wise slice samplers. DEMCMC and iterated Importance samplers are somewhat more robust in this regards; however, both may not scale efficiently to high-dimensional problems. Robustness concerns aside, initial numerical results additionally show some promising further speedups.

Another important branch for future work lies in the utilization of LANs for model comparison. Initial results are promising in that we obtained satisfactory model recovery using the deviance information criterion (DIC) used for model selection in the standard HDDM package. However, this issue demands much more attention to evaluate other model selection metrics and extensive further numerical experiments, which we relegate to future work.

Lastly, in contrast to the importance sampler driving the posterior inference for the CNN, we believe that some of the performance deficiencies of the MLP are the result of our MCs not having converged to the target distribution. A common problem seems to be that the sampler hits the bounds of the constrained parameter space and does not recover from that. As shown in *Figures 7* and *8*, even ostensibly bad parameter recoveries follow a conceptual coherence and lead to good posterior predictive performance. We therefore may be underreporting the performance of the MLP and plan to test the method on an even more comprehensive suite of MCMC samplers, moreover including thus far neglected potential for reparameterization.

## Materials and methods

### Test beds
General information

All models were simulated using the Euler–Maruyama method, which for some fixed discretization step size $\Delta t$ evolves the process as

$$X_{t+\Delta t} = X_t + a(t,x)\Delta t + b(t,x)\Delta \mathbf{B}$$

where the definition of $\Delta \mathbf{B}$ depends on the noise process. For simple Brownian motion, this translates into Gaussian displacements, specifically $\Delta \mathbf{B} \sim \mathcal{N}(0, \Delta t)$, which is commonly denoted as $d\mathbf{W}$. More generally, the noise need not be Gaussian, and indeed we later apply our methods to the Levy flight model for which the noise process is an alpha stable distribution, denoted as $\mathbf{L}_\alpha$ s.t. $\Delta \mathbf{L}_\alpha \sim (\Delta t)^{\frac{1}{\alpha}} \mathbf{L}(\alpha, 0, 1, 0)$.

The models chosen for our test bed systematically vary different aspects of complexity, as illustrated in *Figure 3*. The DDM provides a benchmark and a sanity check since we can compute its likelihood analytically. The full-DDM provides us with a model for which analytical computations are still based on the analytical likelihood of the DDM; however, evaluation is slowed by the necessity for numerical integration. This forms a first test for the speed of evaluation of our methods. For the Ornstein–Uhlenbeck, Levy, Race, and DDM with parameterized boundary models, we cannot base our calculations on an analytical likelihood, but we can nevertheless perform parameter recovery and compare to other methods that utilize empirical likelihoods. The Ornstein–Uhlenbeck model adds state-dependent behavior to the diffusion while the Levy model adds variation in the noise process and the Race models expand the output dimensions according to the number of choices.

## Full-DDM

The full-DDM maintains the same specification for the driving SDE, but also allows for trial-to-trial variability in three parameters (*Ratcliff and McKoon, 2008*). We allow the drift rate $v$ to vary trial by trial; according to a normal distribution, $v \sim \mathcal{N}(0, \sigma_v)$, the nondecision time $\tau$ to vary according to a uniform distribution $\tau \sim \mathbf{U}[-\epsilon_\tau, \epsilon_\tau]$ and the starting point $w$ to vary according to a uniform distribution as well $w \sim \mathbf{U}[-\epsilon_w, \epsilon_w]$. The parameter vector for the full-DDM is then $\theta = (v, a, w, \tau, \sigma_v, \epsilon_\tau, \epsilon_w)$.

To calculate the FPTD for this model, we can use the analytical likelihood expression from the DDM. However, we need to use numerical integration to take into account the random parameters (*Wiecki et al., 2013*). This inflates execution time by a factor equivalent to the number of executions needed to compute the numerical integral.

## Ornstein–Uhlenbeck model

The Ornstein–Uhlenbeck model introduces a state dependency on the drift rate $v$. Here, $a(t, x) = v + g * x$, where $g$ is an inhibition/excitation parameter. If $g < 0$, it acts as a leak (the particle is mean reverting). If $g > 0$, the particle accelerates away from the 0 state, as in an attractor model. At $g = 0$, we recover the simple DDM process. This leaves us with a parameter vector $\theta = (v, a, w, \tau, g)$. The corresponding SDE is defined as

$$d\mathbf{X}_{\tau+t} = (v + g * \mathbf{X}_t)\, dt + d\mathbf{W}, \; \mathbf{X}_\tau = w$$

This model does not have an analytical likelihood function that can be employed for cheap inference (*Mullowney and Iyengar, 2006*). We discuss alternatives, other than our proposed methods, to simple analytical likelihoods later. For our purposes, approximate inference is necessary for this model. The Ornstein–Uhlenbeck model is usually defined only for $g < 0$; our parameter space makes it strictly speaking a relaxation.

## Levy flights

The Levy flight (*Wieschen et al., 2020*; *Reynolds and Rhodes, 2009*) model dispenses with the Gaussian noise assumption in that the incremental noise process instead follows an alpha-stable distribution $\mathcal{L}_\alpha$. Specifically, we consider distributions $\mathcal{L}(\alpha, 0, 1, 0)$ that are centered at 0, symmetric, and have unitary scale parameter. These distributions have a first moment for $\alpha \in (1, 2]$, but infinite variance for $\alpha < 2$. An important special case is $\alpha = 2$, where $\mathcal{L}(2, 0, 1, 0) = \mathcal{N}(0, 2)$. The parameter vector for this process is $\theta = (v, a, w, \tau, \alpha)$. We fix $a(t, x) = v$ and $b(t, x) = 1$. The SDE is defined as

$$d\mathbf{X}_{\tau+t} = v\, dt + d\mathbf{L}_\alpha, \; \mathbf{X}_\tau = w$$

The Levy flight is a flexible model used across disciplines for some of its theoretical optimality properties (*Wosniack et al., 2017*) despite not possessing closed-form FPTDs. We add it here as it is different from the other models under consideration; in principle, it could also capture decision-making scenarios in which there are sudden jumps in the accumulation of evidence (e.g., due to internal changes in attention). Its behavior is shaped by altering the properties of the incremental noise process directly.

## Parameterized collapsing decision bounds

We will consider variations of the DDM in which the decision boundary is not fixed but is time-vary-ing (represented by a boundary parameter $a$ with a parameterized boundary function $h(t; \theta_h)$). In such cases, we augment the parameter vector $\theta$ with the set $\theta_h$ and drop $a$. Such variations are opti-mal in a variety of settings (e.g., when there are response deadlines, *Frazier and Angela, 2008*, or distributions of trial types with different difficulties, *Malhotra et al., 2018*; *Palestro et al., 2018*), and also better reflect the underlying dynamics of decision bounds within biologically inspired neural models (*O'Reilly and Frank, 2006*; *Ratcliff and Frank, 2012*; *Wiecki and Frank, 2013*). The bound-ary functions considered in the following are the Weibull bound (Weibull),

$$b_{WB}(t; a, \alpha, \beta) = a * \exp\left(-\frac{t^\alpha}{\beta}\right)$$

and the linear collapse bound (LC),

$$b_{LC}(t; a, \theta) = a - \left(t * \frac{\sin(\theta)}{\cos(\theta)}\right)$$

## Race models: N > 2

The Race model departs from previous model formulations in that it has a particle for each of $N$ choice options instead of a single particle representing the evidence for one option over another. The function $f_{E_i}(t, \theta)$ now represents the probability of particle $i$ to be the first of all particle to cross the bound $a$ at time $t$. We consider race models for which the drift and starting point can vary for each particle separately. Treating the boundary as a constant $a$ leaves us with a parameter vector $\theta = (v_1, ..., v_n, a, w_1, ..., w_n, ndt)$. The SDE is defined for each particle separately (or in vector form) as

$$d\mathbf{X}_{\tau+t}^i = v^i dt + d\mathbf{W}, \quad \mathbf{X}_0^i = ... = \mathbf{X}_\tau^i = w^i$$

These models represent the most straightforward extension to a multichoice scenario.

## Multilayered perceptron

### Network specifics

We apply the same simple architecture consistently across all example contexts in this paper. Our networks have three hidden layers, $\{L_1, L_2, L_3\}$, of sizes $\{100, 100, 120\}$, each using $tanh(.)$ activation functions. The output layer consists of a single node with linear activation function.

### Training process

#### Training hyperparameters

The network is trained via stochastic back-propagation using the Adam (*Kingma and Ba, 2014*) opti-mization algorithm. As a loss function, we utilize the Huber loss (*Huber, 1992*) defined as

$$f(|y - \hat{y}|) = \begin{cases} 0.5 * |y - \hat{y}|^2 & \text{if } |y - \hat{y}| \leq 1 \\ 0.5 + |y - \hat{y}| & \text{if } |y - \hat{y}| > 1 \end{cases}$$

#### Training data

We used the following approach to generate training data across all examples shown below.

First, we generate 100K simulations from the stochastic simulator (or model $\mathcal{M}$) for each of 1.5 M parameter configurations. Since for the examples we consider the stochasticity underlying the mod-els are in the form of a SDE, all simulations were conducted using the simple Euler–Maruyama method with timesteps $\delta t$ of 0.001 s. The maximum time we allowed the algorithms to run was 20 s, much more than necessary for a normal application of the simulator models under consideration.

Based on these simulations, we then generate empirical likelihood functions using KDEs (*Turner et al., 2015*). KDEs use atomic datapoints $\{x_0, ..., x_N\}$ and reformulate them into a continuous probability distribution $f(y; \mathbf{x}) = \sum_i^N K(\frac{y-x}{h})$, where we choose $K(.)$ as a standard Gaussian kernel $f(x) = \frac{1}{\sqrt{2\pi}} \exp -\frac{x^2}{2}$, and $h$, the so-called bandwidth parameter, is set by utilizing Silverman's rule of thumb (*Silverman, 1986*). Where the data made Silverman's rule inapplicable, we set a lower bound on $h$ as $10^{-3}$. Additionally, we follow *Charpentier and Flachaire, 2015* in transforming our KDE to

accommodate positive random variables with skewed distributions (in adherence to the properties of data resulting from the response time models forming our examples).

To ensure that the networks accurately learn likelihoods across a range of plausible data, for each parameter set we trained the networks by sampling 1000 datapoints from a mixture distribution with three components (mixture probabilities respectively $\{0.8, 0.1, 0.1\}$). The first component draws samples directly from the KDE distributions. The second component is uniform on $[0s, 20s]$, and the third component samples uniformly on $[-1s, 0s]$. The aim of this mixture is to allow the network to see, for each parameter setting of the stochastic simulator, training examples of three kinds: (1) 'Where it matters', that is, where the bulk of the probability mass is given in the generative model. (2) Regions of low probability to inform the likelihood estimate in those regions (i.e., to prevent distortion of likelihood estimates for datapoints that are unlikely to be generated under the model). (3) Examples on the negative real line to ensure that it is learned to consistently drive likelihood predictions to 0 for datapoints close to 0.

The supervision signal for training has two components. For positive datapoints (reaction times in our examples), we evaluate the log-likelihood according to our KDE. Likelihoods of negative datapoints were set to an arbitrary low value of $10^{-29}$ (a log-likelihood of $-66.79$). $10^{-29}$ also served as the lower bounds on likelihood evaluations. While this constrains our accuracy on the very tails of distributions, extremely low evaluations unduly affect the training procedure. Since the generation of training data can easily be parallelized across machines, we simply front-loaded the data generation accordingly. We refer back to *Figure 2* for a conceptual overview.

This procedure yields $1.5B$ labeled training examples on which we train the network. We applied early stopping upon a lack of loss improvement for more than five epochs of training. All models were implemented using Tensorflow (*Abadi et al., 2016*).

We note here that this amount of training examples is likely an overshoot by potentially one or more orders of magnitude. We did not systematically test for the minimum amount of training examples needed to train the networks. Minimal experiments we ran showed that roughly one-tenth of the training examples lead to very much equivalent training results. Systematic minimization of the training data is left for future numerical experiments since we do not deem it essential for purposes of a proof of concept.

## Sampling algorithms

Once trained, we can now run standard MCMC schemes, where, instead of an analytical likelihood, we evaluate $f_w(\mathbf{x}, \theta)$ as a forward pass through the MLP. *Figure 1B* schematically illustrates this approach (following the green arrows) and contrasts with currently applied methods (red arrows). We report multiple so-conducted parameter recovery experiments in the 'Results' section and validate the approach first with models with known analytical likelihood functions.

Regarding sampling, we utilized two MCMC algorithms, which showed generally very similar results. In contrast to the importance sampling algorithm used for the CNN (described below), MCMC methods are known for having trouble with multimodal posteriors. Running our experiments across algorithms was a safeguard against incorporating sampler-specific deficiencies into our analysis. We however acknowledge that even more extensive experiments may be necessary for comprehensive guarantees. First, having an ultimate implementation of our method into the HDDM Python toolbox (*Wiecki et al., 2013*) in view, we use slice sampling (as used by the toolbox), specifically the step-out procedure following *Neal, 2003*. Second, we used a custom implementation of the DEMCMC algorithm (*Braak, 2006*), known for being robust in higher dimensional parameter spaces. Our DEMCMC implementation adds reflecting boundaries to counteract problematic behavior when the sampler attempts to move beyond the parameter space, which is truncated by the (broad) range of parameters in which the MLP was trained. The number of chains we use is consistently determined as $5 * |\theta|$, five times the number of parameters of a given stochastic model. Samplers were initialized by using slight perturbations of five maximum likelihood estimates and computed via differential evolution optimization (*Storn and Price, 1997*; *Virtanen et al., 2020*). Since results were very similar across samplers, we restrict ourselves mostly to reporting results derived from the slice sampler, given that this sampler forms the back-end of the HDDM user interface we envision. Implementations of the MLP method, the samplers we used, as well as the training pipeline can be found at

### Additional notes

Note that we restricted parameter recovery for the MLP to datasets that distributed at least 5% of choices to the less frequently chosen option. This modest filtering accommodates the fact that such datasets were also excluded form the training data for the MLP model since they (1) present difficulties for the KDE estimator and (2) lead to generally less stable parameter estimates (i.e., it is not advisable to use diffusion models when choices are deterministic).

## Convolutional neural network

### Network specifics

The CNN takes as an input a parameter vector θ, giving as output a discrete probability distribution over the relevant dataspace. In the context of our examples below, the output space is of dimensions $\mathcal{R}^{N_c} \times \mathcal{R}^{N_d}$, where $N_c$ is the number of relevant choice alternatives, and $N_d$ is the number of bins for the reaction time for each choice ($N_d = 512$ for all examples below). The network architecture consists of a sequence of three fully connected upsampling layers, $\{L_1^{FC}, L_2^{FC}, L_3^{FC}\}$, of respectively $\{64, 256, 1024\}$ nodes. These are followed by a sequence of three convolutional layers $\{L_1^C, L_2^C, L_3^C\}$ with $1 \times 5$ kernels, and a final fully connected layer with softmax activation. The network size was not minimized through architecture search, which, along with other potential further speed improvements, we leave for future research.

### Training process

For the CNN, we use 100K simulations from the stochastic simulator for each of 3 M parameter vectors and bin the simulation outcomes as normalized counts into $\mathcal{R}^{N_c} \times \mathcal{R}^{N_d}$ slots respectively (looking ahead to our examples, $N_c$ concerns the number of choice outcomes, and $N_d$ the number of bins into which the reaction time outcomes are split for a given simulator). The resultant relative frequency histograms (empirical likelihood functions) $\ell_{empirical}(\theta|\mathbf{x}) \; \forall \theta \in \Theta, \forall x \in \mathcal{X}$, then serve as the target labels during training, with the corresponding parameters θ serving as feature vectors. For a given parameter vector θ, the CNN gives out a histogram $\hat{\ell}_\phi(\theta|\mathbf{x})$, where $\phi$ are the network parameters. The network is then trained by minimizing the KL divergence between observed and generated histograms

$$\mathcal{D}(\hat{\ell}(\theta|\mathbf{x})\|\ell_{empirical}(\theta|\mathbf{x})) = \sum_{i=0}^{c}\sum_{j=0}^{d}\left[\hat{\ell}(\theta|x_{ij})\log\frac{\hat{\ell}(\theta|x_{ij})}{\ell_{empirical}(\theta|x_{ij})}\right]$$

Training $\mathcal{D}(\hat{\ell}(\theta|\mathbf{x})\|\ell_{empirical}(\theta|\mathbf{x}))$ is not the only option. We note that it would have been a valid choice to train on $\mathcal{D}(\ell_{empirical}(\theta|\mathbf{x})\|\hat{\ell}(\theta|\mathbf{x}))$ (*Minka, 2013*) or the symmetrized Kullback–Leibler divergence instead. Training results, however, were good enough for our present purposes to leave a precise performance comparison across those loss functions for future research, leaving room for further improvements.

As for the MLP, we use the Adam optimizer (*Kingma and Ba, 2014*) and implemented the network in Tensorflow (*Abadi et al., 2016*).

### Sampling algorithm

One benefit of using the CNN lies in the enhanced potential for parallel processing across large number of parameter configurations and datapoints. To fully exploit this capability, instead of running a (sequential) MCMC algorithm for our parameter recovery studies, we use iterated importance sampling, which can be done in parallel. Specifically, we use adaptive importance sampling based on mixtures of t-distributions, following a slightly adjusted version of the suggestions in *Cappé et al., 2008*; *Wraith et al., 2009*.

While importance sampling is well established, for clarity and the setting in which we apply it, we explain some of the details here. Importance sampling algorithms are driven by the basic equality

$$\int f(\theta)dx = \frac{f(\theta)}{g(\theta)}g(\theta)d\theta$$

which holds for any pair of probability distributions such that $g(\theta)>0$, where $f(\theta)>0$. $f(\theta)$ is our posterior distribution, and $g(\theta)$ is the proposal distribution. We now sample $N$ tuples $\theta$ according to $g(\theta)$ and assign each $\theta_i$ an importance weight $w_i = \frac{f(\theta_i)}{g(\theta_i)}$.

To get samples from the posterior distribution, we sample with replacement the $\theta$ from the set $\{\theta_0, ..., \theta_n\}$, with probabilities assigned as the normalized weights $\{\tilde{w}_0, ..., \tilde{w}_n\}$. We note that importance sampling is exact for $N \to \infty$. However for finite $N$, the performance is strongly dependent on the quality of the proposal distribution $g(.)$. A bad match of $f(.)$ and $g(.)$ leads to high variance in the importance weights, which drives down performance of the algorithm, as commonly measured by the effective sample size (*Liu, 2008*)

$$\hat{ESS} = \frac{1}{\sum_{n=1}^{N} \bar{w}_n^2}$$

Iterated importance sampling uses consecutive importance sampling rounds to improve the proposal distribution $g(.)$. A final importance sampling round is used to get the importance sample we use as our posterior sample. Specifically, we start with a mixture of t-distributions $g_0(.)$, where $\mathcal{M}$ is the number of mixture components. Each component of $g_0(.)$ is centered at the MAP according to a optimization run (again we used differential evolution). The component-covariance matrix is estimated by a numerical approximation of the Hessian at the respective MAP. Each round $i$, based on the importance sample $\{\mathbf{x}, \mathbf{w}\}_i$, we update the proposal distribution (to a new mixture of t-distributions) using the update equations derived in *Cappé et al., 2008*.

As suggested by *Cappé et al., 2008*, convergence is assessed using the normalized perplexity statistic (the exponentiated Shannon entropy of the importance weights). For run $i$, this is computed as $\exp^{\frac{H^{k,N}}{N}}$, where $H^{k,N} = -\sum_{i=1}^{N} \bar{w}_{k,i} \log \bar{w}_{k,i}$.

To help convergence, we depart from the basic setup suggested in *Cappé et al., 2008* in the following way. We apply an annealing factor $\gamma_k = \max 2^{z-k}, 1 \ z \in \{1, 2, 4, ..., \}$, so that for iteration $k$ of the importance sampler we are operating on the target $f(x)^{\frac{1}{\gamma_k}}$. Smoothing the target during the first iterations helps with successfully adjusting the proposal distribution $g(.)$. *Figure 2* visualizes the CNN approach. Again, we emphasize that more numerical experiments using a larger variety of sampling algorithms are desirable, but are out of the scope for this paper. Implementations of the CNN method, the samplers we used, as well as the training pipeline can be found at https://github.com/lnccbrown/lans/tree/master/al-cnn.

## Strengths and weaknesses

In this section, we clarify a few strengths and weaknesses of the two presented methods and their respective use cases. First, representing the likelihood function datapoint-wise as an MLP output, or globally via the CNN output histogram, affects the potential for parallelization. As exploited by the choice of sampler, the CNN is very amenable to parallelization across parameters since inputs are parameter tuples only. Since the output is represented as a global likelihood histogram, the dataset likelihood is computed as the summation of the elementwise multiplied of bin-log-likelihoods, with a correspondingly binned dataset (counts over bins). This has the highly desirable property of making evaluation cost (time) independent of dataset size. While the MLP in principle allows parallel processing of inputs, the datapoint-wise representation of input values ($\{\theta, x\}$) makes the potential for cross-parameter parallelization dependent on dataset sizes. While a single evaluation of the CNN is more costly, cross-parameter batch processing can make it preferable to the MLP. Second, the CNN has an advantage during training, where the representation of the output as a softmax layer, and corresponding training via minimization of the KL divergence, provides a more robust training signal to ensure probability distributions compared to the purely local one in which the MLP learns a scalar likelihood output as a simple regression problem. Third, and conversely, the MLP formulation is more natural for trial-wise parameter estimates since the histogram representations may be redundant in case datapoints are in fact evaluated one by one (given datapoint-wise parameters induced by trial-by-trial effects on parameter vectors). Give equivalent success in learning likelihoods, we see potential for speedup when using the pointwise approach in this case. In principle, both approaches however allow one to estimate the impact of trial-wise regressors on model parameters during inference, without further training. It is, for example, common in the cognitive neuroscience literature to

allow the cross-trial time-course of EEG, fMRI, or spike signals to be modeled as a trial-by-trial regressor on model parameters of, for example, DDMs (*Wiecki et al., 2013*; *Frank et al., 2015*; *Cavanagh et al., 2011*; *Herz et al., 2016*; *Pedersen and Frank, 2020*). Another relevant example is the incorporation of latent learning dynamics. If a subject's choice behavior is driven by reinforcement learning across stimuli, we can translate this into trial-by-trial effects on the parameter vectors of a generative process model (*Pedersen and Frank, 2020*). These applications are implicitly enabled at no extra cost with the MLP method, while the trial-by-trial split multiplies the necessary computations for the CNN by the number $N$ of datapoints when compared to scenarios that only need dataset-wise parameters. We stress again, however, that in general both the CNN and the MLP can directly be used for hierarchical inference scenarios. The preceding discussion pertains to further potential for optimization and relative strengths, not categorical potential for application to a given scenario. With respect to the latter, both methods are essentially equal.

## Acknowledgements

This work was funded by NIMH grants P50 MH 119467-01 and R01 MH084840-08A1. We thank Michael Shvartsman, Matthew Nassar, and Thomas Serre for helpful comments and discussion regarding the earlier versions of this manuscript. Furthermore, we thank Mads Lund Pederson for help with integrating our methods into the HDDM Python toolbox. Lastly, we would like to thank the two reviewers of the manuscript for helpful suggestions, which improved the readability of the manuscript.

## Additional information

### Competing interests

Michael J Frank: Senior editor, *eLife*. The other authors declare that no competing interests exist.

### Funding

| Funder | Grant reference number | Author |
|---|---|---|
| National Institute of Mental Health | P50 MH119467-01 | Michael J Frank |
| National Institute of Mental Health | R01 MH084840-08A1 | Michael J Frank |

The funders had no role in study design, data collection and interpretation, or the decision to submit the work for publication.

### Author contributions

Alexander Fengler, Conceptualization, Data curation, Software, Formal analysis, Validation, Investigation, Visualization, Methodology, Writing - original draft, Writing - review and editing; Lakshmi N Govindarajan, Conceptualization, Data curation, Software, Formal analysis, Investigation, Visualization, Methodology, Writing - review and editing; Tony Chen, Software, Investigation; Michael J Frank, Conceptualization, Resources, Software, Supervision, Funding acquisition, Validation, Writing - original draft, Writing - review and editing

### Author ORCIDs

Alexander Fengler https://orcid.org/0000-0002-0104-3905
Lakshmi N Govindarajan https://orcid.org/0000-0002-0936-2919
Michael J Frank https://orcid.org/0000-0001-8451-0523

### Decision letter and Author response

Decision letter https://doi.org/10.7554/eLife.65074.sa1
Author response https://doi.org/10.7554/eLife.65074.sa2

## Additional files

**Supplementary files**
- Transparent reporting form

**Data availability**

All code is provided freely and is available at the following links: https://github.com/lnccbrown/lans/tree/master/hddmnn_tutorial, https://github.com/lnccbrown/lans/tree/master/al-mlp and https://github.com/lnccbrown/lans/tree/master/al-cnn (copy archived at https://archive.softwareheritage.org/swh:1:rev:e3369b9df138c75d0e490be0c48c53ded3e3a1d6).

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

# Appendix 1

## Parameter recovery

Here, we provide additional figures concerning parameter recovery studies. *Appendix 1—table 1* summarizes the parameter-wise $R^2$ between ground truth and the posterior mean estimates for each tested model and for each the CNN and MLP (where applicable) methods in turn. For the MLP, results are based on a reference run that used training data constructed from KDE empirical likelihoods utilizing 100k simulations each, and a slice sampler stopped with help of the Geweke diagnostic. Results in the paper are based on slice samplers as well as slice samplers, which explains why not all $R^2$ values match exactly the ones found in other figures. Our findings were, however, generally robust across samplers.

**Appendix 1—table 1.** Parameter recovery for a variety of test bed models.

| DDM | | N | v | a | w | ndt | | |
|---|---|---|---|---|---|---|---|---|
| $R^2$ | MLP | 1024 | 1.0 | 1.0 | 0.99 | 1 | | |
| | | 4096 | 1.0 | 1.0 | 0.99 | 1 | | |
| | CNN | 1024 | 1 | 0.94 | 0.98 | 1 | | |
| | | 4096 | 1 | 1 | 0.99 | 1 | | |
| **DDM-SDV** | | | v | a | w | ndt | sdv | |
| $R^2$ | MLP | 1024 | 0.95 | 0.94 | 0.96 | 1 | 0.57 | |
| | | 4096 | 0.94 | 0.95 | 0.97 | 1 | 0.58 | |
| | CNN | 1024 | 0.98 | 0.97 | 0.98 | 1 | 0.79 | |
| | | 4096 | 0.99 | 0.98 | 0.99 | 1 | 0.87 | |
| **LC** | | | v | a | w | ndt | $\theta$ | |
| $R^2$ | MLP | 1024 | 0.99 | 0.93 | 0.97 | 1 | 0.98 | |
| | | 4096 | 0.99 | 0.94 | 0.98 | 1 | 0.97 | |
| | CNN | 1024 | 0.96 | 0.94 | 0.97 | 1 | 0.97 | |
| | | 4096 | 0.97 | 0.94 | 0.98 | 1 | 0.97 | |
| **OU** | | | v | a | w | ndt | g | |
| $R^2$ | MLP | 1024 | 0.98 | 0.89 | 0.98 | 0.99 | 0.12 | |
| | | 4096 | 0.99 | 0.79 | 0.95 | 0.99 | 0.03 | |
| | CNN | 1024 | 0.99 | 0.94 | 0.97 | 1 | 0.41 | |
| | | 4096 | 0.99 | 0.95 | 0.98 | 1 | 0.45 | |
| **Levy** | | | v | a | w | ndt | $\alpha$ | |
| $R^2$ | MLP | 1024 | 0.96 | 0.94 | 0.84 | 1 | 0.33 | |
| | | 4096 | 0.97 | 0.91 | 0.61 | 1 | 0.2 | |
| | CNN | 1024 | 0.99 | 0.97 | 0.9 | 1 | 0.71 | |
| | | 4096 | 0.99 | 0.98 | 0.95 | 1 | 0.8 | |
| **Weibull** | | | v | a | w | ndt | $\alpha$ | $\beta$ |
| $R^2$ | MLP | 1024 | 0.99 | 0.82 | 0.96 | 1 | 0.2 | 0.43 |
| | | 4096 | 0.99 | 0.8 | 0.98 | 0.99 | 0.26 | 0.41 |
| | CNN | 1024 | 0.98 | 0.91 | 0.96 | 1 | 0.4 | 0.69 |
| | | 4096 | 0.98 | 0.91 | 0.97 | 1 | 0.37 | 0.63 |
| **Full-DDM** | | | v | a | w | ndt | dw | sdv | dndt |
| $R^2$ | MLP | 1024 | 0.95 | 0.94 | 0.88 | 1 | 0 | 0.28 | 0.47 |
| | | 4096 | 0.93 | 0.94 | 0.88 | 1 | 0 | 0.25 | 0.38 |

*Continued on next page*

*Appendix 1—table 1 continued*

| DDM | | N | v | a | w | ndt | | | | | | |
|---|---|---|---|---|---|---|---|---|---|---|---|---|
| | CNN | 1024 | 0.98 | 0.98 | 0.93 | 1 | 0 | 0.62 | 0.79 | | | |
| | | 4096 | 0.99 | 0.99 | 0.97 | 1 | 0 | 0.8 | 0.91 | | | |
| **Race 3** | | | v0 | v1 | v2 | a | w0 | w1 | w2 | ndt | | |
| $R^2$ | CNN | 1024 | 0.88 | 0.86 | 0.89 | 0.19 | 0.49 | 0.51 | 0.5 | 0.99 | | |
| | | 4096 | 0.93 | 0.91 | 0.93 | 0.18 | 0.49 | 0.47 | 0.47 | 1 | | |
| **Race 4** | | | v0 | v1 | v2 | v3 | a | w0 | w1 | w2 | w3 | ndt |
| $R^2$ | CNN | 1024 | 0.73 | 0.68 | 0.71 | 0.73 | 0.11 | 0.49 | 0.5 | 0.48 | 0.49 | 0.99 |
| | | 4096 | 0.79 | 0.76 | 0.77 | 0.81 | 0.18 | 0.5 | 0.5 | 0.51 | 0.55 | 0.99 |
| **LCA 3** | | | v0 | v1 | v2 | a | w0 | w1 | w2 | g | b | ndt |
| $R^2$ | CNN | 1024 | 0.58 | 0.56 | 0.58 | 0.47 | 0.7 | 0.72 | 0.68 | 0.27 | 0.57 | 1 |
| | | 4096 | 0.51 | 0.5 | 0.52 | 0.44 | 0.67 | 0.67 | 0.66 | 0.23 | 0.52 | 1 |
| **LCA 4** | | | v0 | v1 | v2 | v3 | a | w0 | w1 | w2 | w3 | g | b | ndt |
| $R^2$ | CNN | 1024 | 0.5 | 0.46 | 0.54 | 0.51 | 0.51 | 0.71 | 0.69 | 0.69 | 0.67 | 0.18 | 0.7 | 0.99 |
| | | 4096 | 0.42 | 0.42 | 0.46 | 0.42 | 0.52 | 0.67 | 0.63 | 0.68 | 0.65 | 0.15 | 0.64 | 1 |

MLP: multilayered perceptron; CNN: convolutional neural network; DDM: drift diffusion model; LC: linear collapse; LCA: leaky competing accumulator.

## Manifolds/likelihoods

We show some examples of the likelihood manifolds for the various models that we tested.

### DDM-SDV

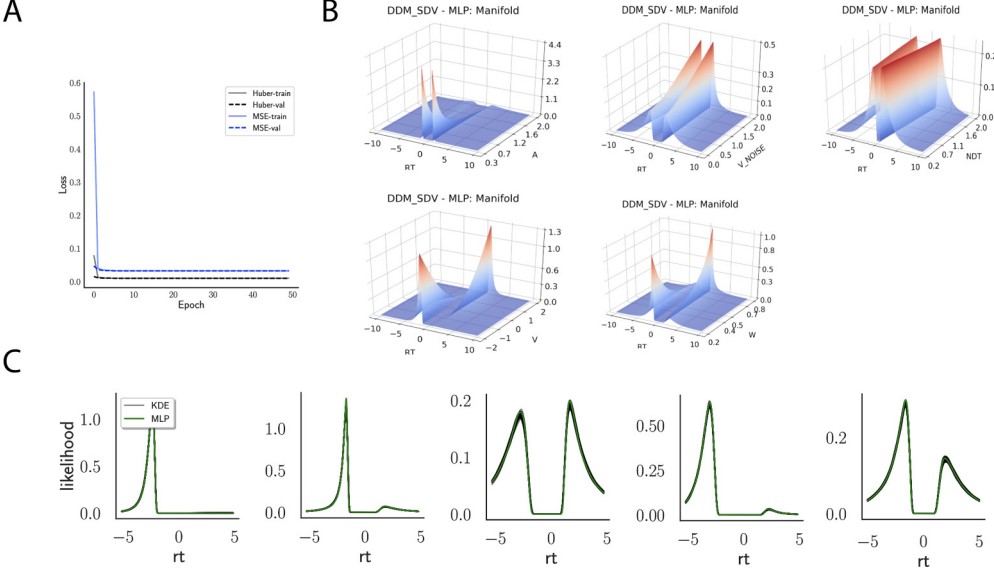

**Appendix 1—figure 1.** Likelihoods and manifolds: DDM-SDV. (**A**) shows the training and validation loss for Huber as well as mean squared error (MSE) for the drift diffusion model (DDM)-SDV model. Training was driven by the Huber loss. (**B**) illustrates the likelihood manifolds by varying one parameter in the trained region. (**C**) shows multilayered perceptron likelihoods in green on top of a sample of 50 kernel density estimate-based empirical likelihoods derived from 20k samples each.

## Linear collapse

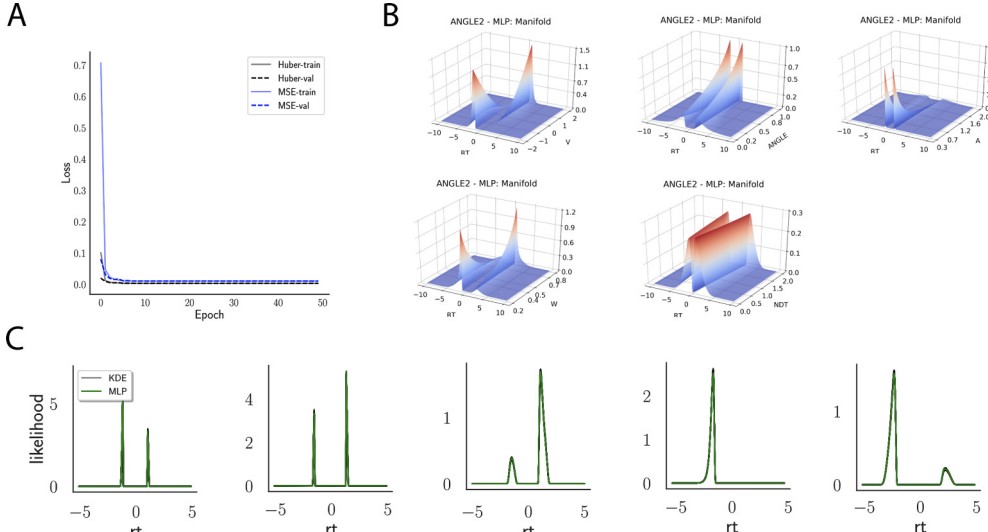

**Appendix 1—figure 2.** Likelihoods and manifolds: linear collapse. (**A**) shows the training and validation loss for Huber as well as MSE for the linear collapse model. Training was driven by the Huber loss. (**B**) illustrates the likelihood manifolds by varying one parameter in the trained region. (**C**) shows multilayered perceptron likelihoods in green on top of a sample of 50 kernel density estimate-based empirical likelihoods derived from 20k samples each.

## Weibull

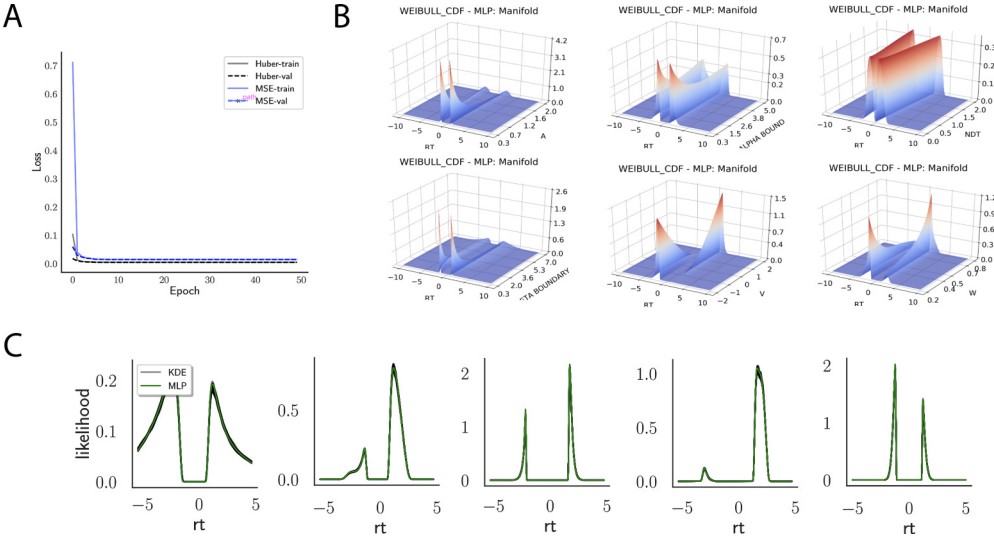

**Appendix 1—figure 3.** Likelihoods and manifolds: Weibull. (**A**) shows the training and validation loss for Huber as well as MSE for the Weibull model. Training was driven by the Huber loss. (**B**) illustrates the likelihood manifolds by varying one parameter in the trained region. (**C**) shows multilayered perceptron likelihoods in green on top of a sample of 50 kernel density estimate-based empirical likelihoods derived from 100k samples each.

## Levy

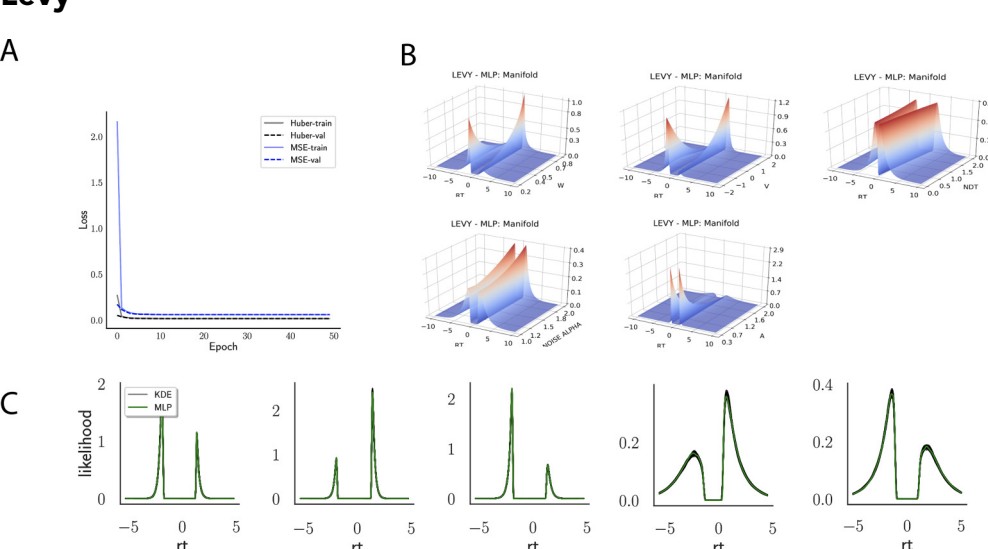

**Appendix 1—figure 4.** Likelihoods and Manifolds: Levy. (**A**) shows the training and validation loss for Huber as well as MSE for the Levy model. Training was driven by the Huber loss. (**B**) illustrates the likelihood manifolds by varying one parameter in the trained region. (**C**) shows multilayered perceptron likelihoods in green on top of a sample of 50 kernel density estimate-based empirical likelihoods derived from 100k samples each.

## Ornstein

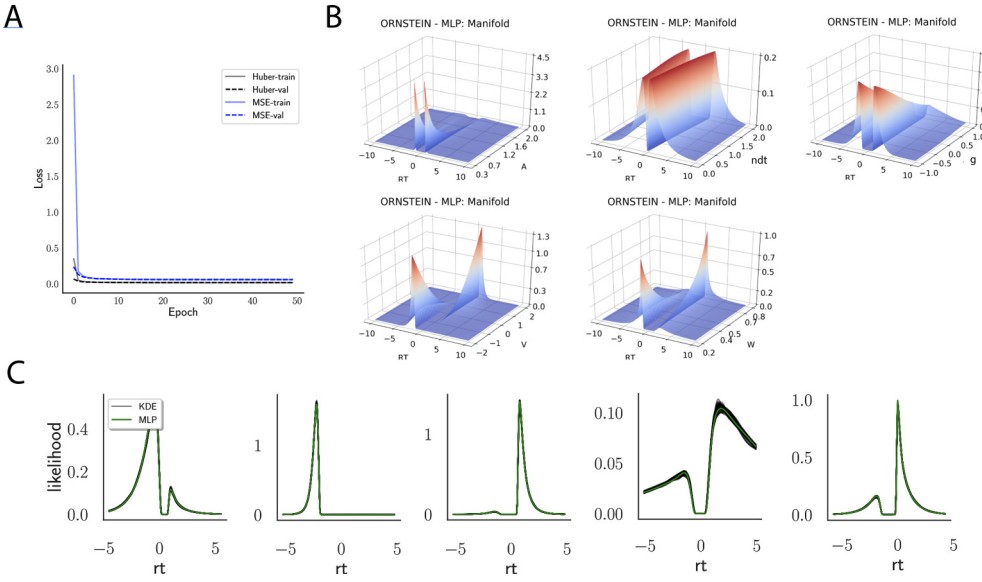

**Appendix 1—figure 5.** Likelihoods and manifolds: Ornstein. (**A**) shows the training and validation loss for Huber as well as MSE for the Ornstein model. Training was driven by the Huber loss. (**B**) illustrates the likelihood manifolds by varying one parameter in the trained region. (**C**) shows multilayered perceptron likelihoods in green on top of a sample of 50 kernel density estimate-based empirical likelihoods derived from 100k samples each.

## Full-DDM

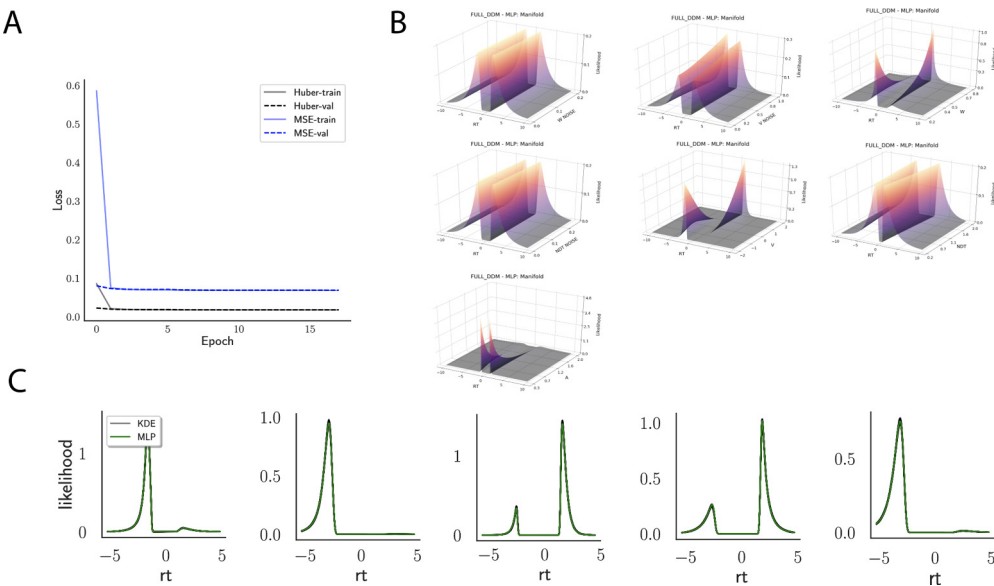

**Appendix 1—figure 6.** Likelihoods and manifolds: full-DDM. (**A**) shows the training and validation loss for Huber as well as MSE for the full drift diffusion model. Training was driven by the Huber loss. (**B**) illustrates the likelihood manifolds by varying one parameter in the trained region. (**C**) shows multilayered perceptron likelihoods in green on top of a sample of 50 kernel density estimate-based empirical likelihoods derived from 100k samples each.

