## [Decision Letter]

**Acceptance summary:**

Your work addresses a very timely and important issue in the field: the restricted availability of candidate computational models that can be tested as explanations of brain or behavioral data. Your approach, which uses artificial neural networks for speeding up inference, helps achieving significantly faster solutions for fitting computational models to data, with potential impact for a broad research community. The ability to decouple the slow training of neural networks to learn the likelihood function of a given model from the fast use of previously trained neural networks to fit data is particularly appealing. We congratulate you for this work which should stimulate the exploration of previously untested computational models of brain activity and behavior.

**Decision letter after peer review:**

Thank you for submitting your article "Likelihood Approximation Networks (LANs) for Fast Inference of Simulation Models in Cognitive Neuroscience" for consideration by *eLife*. Your article has been reviewed by 2 peer reviewers, and the evaluation has been overseen by Valentin Wyart as the Reviewing Editor and Timothy Behrens as the Senior Editor. The following individuals involved in review of your submission have agreed to reveal their identity: Luigi Acerbi (Reviewer #1); Jean Daunizeau (Reviewer #2).

The reviewers have discussed the reviews with one another and the Reviewing Editor has drafted this decision to help you prepare a revised submission.

This manuscript describes a novel method based on artificial neural networks to approximate the trial likelihood function of a model whose likelihood is difficult or impossible to evaluate analytically. The authors demonstrate their method on a class of generalized drift-diffusion models, performing Bayesian inference and various forms of hierarchical inference, with good empirical results. The authors discuss the appeal of their method for broadening the family of models that can be employed in computational cognitive neuroscience.

Both reviewers have found your manuscript to be addressing a very timely and important issue in the field, namely: the restricted availability of candidate computational models that can be tested as explanations of observed data. The described work fits very well with recent work in the area which uses (deep) neural networks for speeding up inference. Both reviewers have found that the methods proposed in the manuscript can indeed help achieve faster solutions when fitting computational models to data, with potential impact for the scientific community. Both reviewers felt that the manuscript deserves publication in *eLife* given that the following points outlined below are addressed to improve the clarity of the presentation and also the limits of the proposed method.

Essential revisions:

1. Clarifications:

For most of the manuscript, the authors make a distinction of "analytical likelihood" vs. "no likelihood" (where one needs to do simulation-based inference). However, there is a middle-ground of models for which the likelihood is not analytical but can be computed numerically. These models constitute a superset of the analytical models. The authors touch upon this point only in the Discussion (when talking about Shinn et al. 2020 and generalized DDMs), but this point and the reference should come up much earlier as it is quite important. There are still good reasons for wanting to do amortized inference as opposed to using numerical likelihoods (the latter can be very expensive, and they don't exist for all DDMs), but the readers should be made aware of these alternative options – even beyond the DDM example.

Mathematical notations are problematic. First, most notations are not described when they first appear, and the reader often has to guess what the variables are. For example, the KDE notation is first presented when introducing the ABC approach (section 2), but one has to read the Additional Methods (section 10) to find out what they really are. Second, mathematical notations are not fully consistent throughout the paper. For example, the parameters are called either P or theta, the data is sometimes called D and sometimes X, whereas X also sometimes denotes the DDM's accumulated evidence. The variable W sometimes means the ANN's weights, sometimes the DDM's Wiener noise, sometimes the DDM's initial condition. The authors should make sure that notations are self-contained and consistent throughout the manuscript.

Another concern regarding mathematical notations is that the typical notation for modelling likelihoods in cognitive science is p(x|s,theta), where x is the observation/response, s a vector describing the "stimulus" in the current trial, and theta the model parameters. In this paper, the authors merge s into theta, which is fine, but it is important to specify it (e.g., this affects the dimensionality of theta).

Important aspects of the training of the networks should be explained earlier in the manuscript. First, the networks are trained on hundreds of billions of simulations: O(10^6^) parameter settings times 10^5^ samples per parameter setting. This ballpark should be mentioned early (e.g., in Figure 1 and 2 or related text). More generally, it is important to stress that amortized inference is not free at all – it incurs a very large cost upfront. So, for example, it is likely not worth it in the scenario in which a researcher is iterating through many different models (as opposed to a single model applied many times). Second, the initial introduction of the neural network models (lines 255-269) is confusing, due to some unfortunate choice of wording in the explanation (and despite the fact that what the authors are doing is quite simple):

1) In the pointwise representation, the network is learning a mapping from x and theta to p(x|theta).

2) In the histogram representation, the network is learning a mapping from theta to p(x|theta), where p(x|theta) is represented as a histogram over the space of (discretized) observations x.

What is confusing is that the choice of architecture used in the paper (MLP for pointwise, CNN for histogram) seems to be orthogonal to what the network is representing. The histogram representation could use a MLP architecture. In fact, the authors should explain why they state that the histogram representation "naturally leads to CNNs as the network architecture" (e.g., there seems to be no translational invariance in the problem). Why is MLP a priori better for approximating pointwise likelihoods, and CNN better for histogram likelihoods? What, in these architectures, makes them suited for the tasks? For example, what would authors expect, if they used CNNs for pointwise likelihoods and MLPs for histogram likelihoods? Did authors try different architectures and ended concluding that this was the best choice, or is it an educated guess?

Also, the expression "discretized global likelihood of an entire dataset" confusing, because to me that would be learning p(D|theta). Instead, the authors mean learning the full trial likelihood function p(x|theta), which in their case helps compute the likelihood of the entire datasets (because the trials are assumed to be i.i.d.). Last, the authors use "parameterization" when they mean "parameter vector".

Overall, it would be very helpful if the authors were to rewrite this part more clearly. Figure 2 could be made larger, highlighting the input and output of the networks.

2. Limitations of the method:

Related to an earlier point, it is important to stress explicitly that amortized inference is not free at all – it incurs a very large cost upfront (here, training ANNs on hundreds of billions of simulations: O(10^6^) parameter settings times 10^5^ samples per parameter setting). So, for example, it is likely not worth it in the scenario in which a researcher is iterating through many different models (as opposed to a single model applied many times).

The method described only applies to models whose likelihood is conditionally independent given the model and trial parameters (theta); that is whose likelihood factorizes as p(D|theta) = \prod_i_ p(x_i_|theta). Models with a Markovian dependence on previous observations can still be addressed (by putting a limited number of previous observations into theta). However, the likelihood above does not apply to models with latent dynamics. The models that satisfy the requirements of the method are still plenty of models in cognitive science, but still, it is very important to specify the restrictions of the method. The authors can look at the section "Definitions and notation" of van Opheusden et al. (2020) for a more in-depth discussion.

The authors should state clearly the scalability of the proposed approach based on the dimensionality of theta (which includes the "stimulus" s, see an earlier point). The bottleneck here seems to be the coverage of the "prior" space. It is already impressive that the method works alright in 6-7 dimensions; still, it can be expected to worsen quickly due to the curse of dimensionality (not because of the network – but because of the inability of the prior sampling to cover the space). For example (which would be good to mention), Goncalves et al. (2020) have shown that neural networks can reconstruct posteriors in up to ~30 dimensions (with some tricks). However, to do so they cannot use pure, "a priori" amortized inference, but they need to iterate the sampling procedure to zoom into the region of interest.

Relatedly, the authors might want to comment on what happens when s or theta is not a continuous parameter (for example, see again van Opheusden et al., 2020; in one of the models, s takes an integer value of a "board configuration" which is not necessarily part of a nice metric space; e.g. board state 15 is not necessarily close to 16). Clearly the network can still work in principle but effectively we would expect discontinuities (due to learning in a discrete space) which might cause issues. In other words, unless the authors have some evidence that their method also works well for discrete parameters, they might want to mention this limitation of the method.

Relatedly also, the authors should explain more clearly how they chose the range of parameters which they sample from during ANN training. In a sense, and although the goal here is to approximate the likelihood, it seems to me that this step effectively places some form of implicit prior distribution on parameters (at least in terms of admissible parameter values). In turn, this induces some estimation artefacts, which can be eyeballed on some panels of Figure 8 (cf. α and β bound parameters of the Weibull model). Wouldn't the estimation error be much higher, had the authors simulated data under broader parameter ranges (further beyond the sampling range that was used for ANN training)? Does a restricted sampling range not eventually induce some form of potential non-identifiability issue? These points should be discussed more explicitly in the manuscript.

The authors have shown that the method they propose yields efficient (Bayesian) statistical inference on model parameters. However, one may also want to use the method for comparing models. Can authors demonstrate the model comparison capabilities of their method? If the authors believe this is out of scope, they should at least discuss this possible application of their method. ABC methods have sometimes been criticized for the fact that they induce model section biases. However, these biases may be related to the selection of data sufficient statistics (which may be unfair to some models). How do these points translate to the current approach?

The authors have demonstrated the appeal of their method on DDM variants, which typically predict response times and choices. Practically speaking, this means storing the trained ANNs for these two types of dependent variables. But what if one uses these models to predict another set of dependent behavioral variables (e.g. including choice confidence)? It seems that one would have to re-perform all the simulations. Or is there a way to leverage what has already been done? The authors may want to discuss this point somewhere in the manuscript.

3. Figures are not described with a sufficient amount of detail, and the read has at times to guess what captions do not state. For example, what do negative RT mean in Figure 4 (I guess: RT conditional on "the other" choice outcome)? What are the middle panels of Figure 7 showing (I guess: posterior intervals on ANGLE bounds over decision time, as well as empirical and fitted choice-dependant RT histograms)? What are the X-1024 and X-4096 variants of their ANN-based approach? What are the nodes/variables in the DAGs of Figure 10? These ambiguities should be solved in the revised manuscript.

4. The relationship of this work to a few related lines of work may deserve to be discussed. In particular, the approach proposed here appears to share similarities to (variational) autoencoders, which is now an established ANN-based method for solving very similar problems. Also, it may be worth discussing the pros and cons of the authors' approach, when compared to other existing approaches to likelihood-free inference, which would approximate the likelihood in some other way (using, e.g., Gaussian processes). Anecdotally, for the specific case of 'vanilla' DDM, there are recent advances to the problem of deriving conditional and marginal moments of RT distributions (Srivastava et al., 2016). These are much faster to evaluate than Fuss and Navarro (2009), because they provide analytical expressions that do not require any iterative numerical computations. Embedded within an ad-hoc likelihood (derived from an observation equation of the form: RT = E[RT|theta] + i.i.d. Gaussian noise), these can provide very fast/efficient parameter estimates. It would be worth discussing these kinds of approaches somewhere in the Discussion section.

---

## [Author Response]

Essential revisions:1. Clarifications:For most of the manuscript, the authors make a distinction of "analytical likelihood" vs. "no likelihood" (where one needs to do simulation-based inference). However, there is a middle-ground of models for which the likelihood is not analytical but can be computed numerically. These models constitute a superset of the analytical models. The authors touch upon this point only in the Discussion (when talking about Shinn et al. 2020 and generalized DDMs), but this point and the reference should come up much earlier as it is quite important. There are still good reasons for wanting to do amortized inference as opposed to using numerical likelihoods (the latter can be very expensive, and they don't exist for all DDMs), but the readers should be made aware of these alternative options – even beyond the DDM example.

We agree that this is an important distinction that the paper should have emphasized more, beyond alluding to this in Figure 3 (whereby the full DDM was depicted in a different color because it requires numerical integration and is therefore costly to evaluate). We now make it explicit in Section 3 and mention it again in the main text in Section 4 when describing the test-bed models. Indeed, powerful numerical approximation methods exist for some likelihoods, but these do not afford straightforward or rapid Bayesian inference. An example of this is already present in the full DDM, given the numerical integration, which typically slows down estimation compared to standard DDM by at least an order of magnitude. Informal tests against the full DDM likelihood in the HDDM python package (Wickie et al. 2013) show that we gain an approximately 40 fold increase in speed for this model when using LANs instead. Notably, the generalized DDMs discussed in Shinn et al. 2020 did not include a method for posterior estimation of parameters (which would likely be quite costly to compute). Moreover, these generalized DDMs are limited to cases with Gaussian noise. Any departures from that, such as Levy flights, are not accommodated. This issue is not a concern for LANs.

Mathematical notations are problematic. First, most notations are not described when they first appear, and the reader often has to guess what the variables are. For example, the KDE notation is first presented when introducing the ABC approach (section 2), but one has to read the Additional Methods (section 10) to find out what they really are. Second, mathematical notations are not fully consistent throughout the paper. For example, the parameters are called either P or theta, the data is sometimes called D and sometimes X, whereas X also sometimes denotes the DDM's accumulated evidence. The variable W sometimes means the ANN's weights, sometimes the DDM's Wiener noise, sometimes the DDM's initial condition. The authors should make sure that notations are self-contained and consistent throughout the manuscript.Another concern regarding mathematical notations is that the typical notation for modelling likelihoods in cognitive science is p(x|s,theta), where x is the observation/response, s a vector describing the "stimulus" in the current trial, and theta the model parameters. In this paper, the authors merge s into theta, which is fine, but it is important to specify it (e.g., this affects the dimensionality of theta).

We thank the reviewers for pointing this out. We have now improved notational consistencies. We now refer to network parameters as Φ, and datasets and data points are consistently 𝐱 and 𝑥. Model parameters and parameter dimensions are now always 𝜃 and |_Θ|_. Wiener noise is mentioned once as a capital bold 𝑑_𝐖_, which is distinct from (lower case) 𝑤 as the starting point parameter. We also made explicit that we suppress the stimulus 𝑠 in our discussions when first defining the likelihood.

Important aspects of the training of the networks should be explained earlier in the manuscript. First, the networks are trained on hundreds of billions of simulations: O(10^6^) parameter settings times 10^5^ samples per parameter setting. This ballpark should be mentioned early (e.g., in Figure 1 and 2 or related text). More generally, it is important to stress that amortized inference is not free at all – it incurs a very large cost upfront. So, for example, it is likely not worth it in the scenario in which a researcher is iterating through many different models (as opposed to a single model applied many times).

We have now added additional details to our initial discussion of the ANNs and training procedures to help readability. As we now reiterate in the text, our approach’s primary benefit lies in reusability and flexibility, which we believe will afford practical advantages for end users to evaluate models of interest across arbitrary inference scenarios. Note that we did not attempt a structured minimization of training data. Indeed, informal experiments indicate performance would be quite similar with an order of magnitude less training. The reason we are not so focused on the precise training time is that the cost of training LANs is only incurred once for any theoretically meaningful model. At this point, the trained network can be added to a bank of LANs, which can be made available to the community and reused without further training. In turn, the community will be able to test their data against a broader variety of models (and to easily add their own to the bank) and which can be evaluated for arbitrary inference scenarios (hierarchical estimation, allow some parameters to vary across conditions or to relate to neural measure X, etc.). While it is true that, in principle, we would prefer to further minimize amortization cost (and we may attempt to do so in future), we view in the long run that this component is comparatively negligible for any model that would be reused.

Second, the initial introduction of the neural network models (lines 255-269) is confusing, due to some unfortunate choice of wording in the explanation (and despite the fact that what the authors are doing is quite simple):1) In the pointwise representation, the network is learning a mapping from x and theta to p(x|theta).2) In the histogram representation, the network is learning a mapping from theta to p(x|theta), where p(x|theta) is represented as a histogram over the space of (discretized) observations x.What is confusing is that the choice of architecture used in the paper (MLP for pointwise, CNN for histogram) seems to be orthogonal to what the network is representing. The histogram representation could use a MLP architecture. In fact, the authors should explain why they state that the histogram representation "naturally leads to CNNs as the network architecture" (e.g., there seems to be no translational invariance in the problem). Why is MLP a priori better for approximating pointwise likelihoods, and CNN better for histogram likelihoods? What, in these architectures, makes them suited for the tasks? For example, what would authors expect, if they used CNNs for pointwise likelihoods and MLPs for histogram likelihoods? Did authors try different architectures and ended concluding that this was the best choice, or is it an educated guess?

We agree that the proposed network architectures are not hard constraints, and we did not intend to imply otherwise. Indeed, we had discussed this very issue amongst ourselves. We have now clarified in the manuscript that the problem representation is what matters. We pursued the architectures in each case that seemed to follow naturally but without any commitment to their optimality. When learning a mapping from simulation model parameters to a histogram representation of the likelihood, we map a low dimensional data manifold to a much higher dimensional one. Using an MLP for this purpose would imply that the number of neural network parameters needed to learn this function would be orders of magnitude larger than using a CNN. Not only would this mean that “forward passes“ through the network would take longer, but also an increased propensity to overfit on an identical data budget. Given that our networks performed adequately well in the training objective and parameter recovery, we felt that we did not need to pursue a rigorous architecture search for this paper’s main points to be developed. We will optimize these for speed in future work.

Also, the expression "discretized global likelihood of an entire dataset" confusing, because to me that would be learning p(D|theta). Instead, the authors mean learning the full trial likelihood function p(x|theta), which in their case helps compute the likelihood of the entire datasets (because the trials are assumed to be i.i.d.).

We recognized this as potentially confusing and rewrote sections of the related paragraphs for clarity.

Last, the authors use "parameterization" when they mean "parameter vector".Overall, it would be very helpful if the authors were to rewrite this part more clearly.

Thank you, we have now fixed in response to the same point elsewhere in the reviewer comments.

Figure 2 could be made larger, highlighting the input and output of the networks.

Thanks for this suggestion. We have now edited Figure 2 to showcase the neural networks more prominently.

2. Limitations of the method:Related to an earlier point, it is important to stress explicitly that amortized inference is not free at all – it incurs a very large cost upfront (here, training ANNs on hundreds of billions of simulations: O(10^6^) parameter settings times 10^5^ samples per parameter setting). So, for example, it is likely not worth it in the scenario in which a researcher is iterating through many different models (as opposed to a single model applied many times).

Please see a detailed response to this comment above. We now added these aspects at the end of section 3, and included some discussion that puts the quoted numbers into perspective. We agree that this will likely aid readability and help with a more nuanced perspective on global amortization.

The method described only applies to models whose likelihood is conditionally independent given the model and trial parameters (theta); that is whose likelihood factorizes as p(D|theta) = \prod_i_ p(x_i_|theta). Models with a Markovian dependence on previous observations can still be addressed (by putting a limited number of previous observations into theta). However, the likelihood above does not apply to models with latent dynamics. The models that satisfy the requirements of the method are still plenty of models in cognitive science, but still, it is very important to specify the restrictions of the method. The authors can look at the section "Definitions and notation" of van Opheusden et al. (2020) for a more in-depth discussion.

We thank the reviewers for raising this point. The problems with amortization when one of the “parameters“ may be a discrete space of high cardinality are indeed real. However, in general our approach can in fact incorporate latent processes on the parameters, which can induce sequential dependence. An example is the RL-DDM model in Pedersen et al. 2020, which puts a latent reinforcement learning process on some of the DDM parameters, which we can easily accommodate with LANs, and which we note in the discussion.

The authors should state clearly the scalability of the proposed approach based on the dimensionality of theta (which includes the "stimulus" s, see an earlier point). The bottleneck here seems to be the coverage of the "prior" space. It is already impressive that the method works alright in 6-7 dimensions; still, it can be expected to worsen quickly due to the curse of dimensionality (not because of the network – but because of the inability of the prior sampling to cover the space). For example (which would be good to mention), Goncalves et al. (2020) have shown that neural networks can reconstruct posteriors in up to ~30 dimensions (with some tricks). However, to do so they cannot use pure, "a priori" amortized inference, but they need to iterate the sampling procedure to zoom into the region of interest.

Indeed, we would generally not advise to apply amortization with LANs naively for models of > _15_ parameters (the CNN was tested on models in that ballpark with the LCA4 in the paper) – or at least we haven’t tested that scenario yet. It is correct that in general the curse of dimensionality can hardly be avoided here, if global amortization is the goal. We do cite Goncalves et al. (2020) twice, and we now also explicitly mention the ballpark figure of 30 parameters with their method. However, a major part of our argument is that modularity of the amortized piece of the pipeline is important. While the amortization of a 30 dimensional posterior is impressive, it remains inflexible to any consideration of experimental design and in spirit focuses on ’single model’, ’single dataset’ fits. Even hierarchical estimation of the basic DDM model with 10 subjects, will elevate us to > 40 parameters and render posterior amortization futile. While we emphasize the virtues of our approach, which lies in the flexibility and reuse, we certainly agree that it is not always the right choice to amortize likelihoods and that in some situations one may rightfully prefer the methods of e.g. Goncalves et al. (2020) (and others). Scaling of our approach with respect to a single model of possibly large parameter spaces, should follow the limitations of the approach of Goncalves et al. (2020), when training is restricted to one round (without ’zooming in’).

Relatedly, the authors might want to comment on what happens when s or theta is not a continuous parameter (for example, see again van Opheusden et al., 2020; in one of the models, s takes an integer value of a "board configuration" which is not necessarily part of a nice metric space; e.g. board state 15 is not necessarily close to 16). Clearly the network can still work in principle but effectively we would expect discontinuities (due to learning in a discrete space) which might cause issues. In other words, unless the authors have some evidence that their method also works well for discrete parameters, they might want to mention this limitation of the method.

Thank you, we have added this limitation briefly. In principle discrete parameters can work, however in general it is true that it may not be advisable for parameters which have a lot of discrete states, for which conditioning on the positions on a chess-board is a good example. The resulting manifold will likely not be smooth enough. Also, the amount of training data needed to amortize such scenario fully, may simply render global amortization an unreasonable computational strategy to begin with.

Relatedly also, the authors should explain more clearly how they chose the range of parameters which they sample from during ANN training. In a sense, and although the goal here is to approximate the likelihood, it seems to me that this step effectively places some form of implicit prior distribution on parameters (at least in terms of admissible parameter values). In turn, this induces some estimation artefacts, which can be eyeballed on some panels of Figure 8 (cf. α and β bound parameters of the Weibull model). Wouldn't the estimation error be much higher, had the authors simulated data under broader parameter ranges (further beyond the sampling range that was used for ANN training)? Does a restricted sampling range not eventually induce some form of potential non-identifiability issue? These points should be discussed more explicitly in the manuscript.

Generally performance will not be optimal if the generative parameters are too close (or outside as one may consider for experimental data) to the boundary of the training region, or if a dataset is defective in a number of ways (e.g., too few choices in one or the other category) which reduces parameter identifiability even for analytic likelihood methods. We now added some clarifying remarks at the end of the section on parameter recovery. The parameter spaces were chosen so as to strongly envelope reasonable expectations for experimental data. In actuality, our LANs were trained to cover a far broader range of RTs compared to the general scope in which DDM-like models are evaluated (usually decisions that take a few seconds; our LANs cover up to 30 seconds, which could always be further expanded but is unlikely to be meaningful). We explain this again in a response to a similar criticism below. We discuss this as well in response to the comment about Line 438-489.

The authors have shown that the method they propose yields efficient (Bayesian) statistical inference on model parameters. However, one may also want to use the method for comparing models. Can authors demonstrate the model comparison capabilities of their method? If the authors believe this is out of scope, they should at least discuss this possible application of their method. ABC methods have sometimes been criticized for the fact that they induce model section biases. However, these biases may be related to the selection of data sufficient statistics (which may be unfair to some models). How do these points translate to the current approach?

We now discuss this in section 8. It is indeed an important topic and a feasible application of the method. Initial results in the context of an extension to the HDDM package, show that DIC appears to work adequately to the same degree it does for analytic likelihood. However, to properly develop the model comparison aspect, and to do it justice would demand a substantial addition to the paper, which we indeed intended to leave for future work.

The authors have demonstrated the appeal of their method on DDM variants, which typically predict response times and choices. Practically speaking, this means storing the trained ANNs for these two types of dependent variables. But what if one uses these models to predict another set of dependent behavioral variables (e.g. including choice confidence)? It seems that one would have to re-perform all the simulations. Or is there a way to leverage what has already been done? The authors may want to discuss this point somewhere in the manuscript.

Thanks for this point. While there are some models that consider confidence as arising directly from the decision variable in DDMs (e.g., van den Berg et al., 2016), here we conceptualize this question more generally as asking about an approach to a joint model, where confidence is modeled conditioned on reaction times and choices, with some probabilistic model 𝑝_𝑀𝑐𝑜𝑛𝑓_ (𝑐𝑜𝑛𝑓𝑖𝑑𝑒𝑛𝑐𝑒|𝑟𝑡, 𝑐, 𝜃_𝑐𝑜𝑛𝑓_ ).

As a whole we would be left with,

𝑝𝑀_𝑐𝑜𝑛𝑓_ (𝑐𝑜𝑛𝑓𝑖𝑑𝑒𝑛𝑐𝑒|𝑟𝑡, 𝑐, 𝜃_𝑐𝑜𝑛𝑓_ )𝑝_𝐷𝐷𝑀_(𝑟𝑡, 𝑐|𝜃_𝐷𝐷𝑀_)

In essence, the question becomes how to “integrate out“ reaction times and choices from this model of choice confidence. This can be achieved either by using numerical integration using pre-computed LANs, or by attempting to train the joint model directly as a LAN (in principle training data from the DDM-LAN could be reused here or the DDM-LAN itself, a variety of options would exist). This is, in fact, an important second-order application, which we hope to address in future work.

3. Figures are not described with a sufficient amount of detail, and the read has at times to guess what captions do not state. For example, what do negative RT mean in Figure 4 (I guess: RT conditional on "the other" choice outcome)? What are the middle panels of Figure 7 showing (I guess: posterior intervals on ANGLE bounds over decision time, as well as empirical and fitted choice-dependant RT histograms)? What are the X-1024 and X-4096 variants of their ANN-based approach? What are the nodes/variables in the DAGs of Figure 10? These ambiguities should be solved in the revised manuscript.

Thank you for pointing out the lack of clarity in the captions. We hope to have resolved the resulting ambiguities. Figure 10 now explains the plate notation used and what 𝜃 and 𝜆 stand for right away. We added explanations regarding the legends in Figure 9. We added information to Figure 7 so that we explicitly describe the left, middle and right panels. We clarified what negative RTs mean in Figure 4 (indeed they are mirrored to display RTs for the alternative choice).

4. The relationship of this work to a few related lines of work may deserve to be discussed. In particular, the approach proposed here appears to share similarities to (variational) autoencoders, which is now an established ANN-based method for solving very similar problems. Also, it may be worth discussing the pros and cons of the authors' approach, when compared to other existing approaches to likelihood-free inference, which would approximate the likelihood in some other way (using, e.g., Gaussian processes). Anecdotally, for the specific case of 'vanilla' DDM, there are recent advances to the problem of deriving conditional and marginal moments of RT distributions (Srivastava et al., 2016). These are much faster to evaluate than Fuss and Navarro (2009), because they provide analytical expressions that do not require any iterative numerical computations. Embedded within an ad-hoc likelihood (derived from an observation equation of the form: RT = E[RT|theta] + i.i.d. Gaussian noise), these can provide very fast/efficient parameter estimates. It would be worth discussing these kinds of approaches somewhere in the Discussion section.

1. Variational Autoencoder: While a link can be drawn to autoencoders (see Fengler et al.(2020) Cognitive Science Society conference paper), we consider it not essential and potentially misleading (this reflects an update in our own thinking), in the following sense. Our networks are not attempting to learn latent variables, but rather the ’forward part’ of the problem. We ultimately use the ’forward part’ to infer the latent parameter vectors 𝜃 as part of the Bayesian inference framework, but this has little to do with the amortization strategy. We are not attempting a mapping from data back to data (via a bottleneck), in both strategies and moreover the learned mappings are not stochastic. Hence we decided to leave out any connection to the variational autoencoder framework.

2. Gaussian Processes: We rewrote parts of sections 3 and expanded the discussion on related works to include Gaussian Process approaches and other likelihood based variants, to make sure the contextualization is more thorough early on, instead of relying solely on elaborations in the discussion.

3. ’Vanilla’ DDM alternatives: The approach of Srivastava et al. (2016) aims to capture moments of RT distributions. While one could build adhoc likelihood functions of the form suggested by the reviewer, this seems to defeat the purpose of attempting to be faithful to the actual likelihood (there seems to be no guarantee that basing the likelihood on 𝑅𝑇 = 𝐸[𝑅𝑇|𝑡ℎ𝑒𝑡𝑎]+𝑒𝑝𝑠), which may lead to highly inaccurate posteriors when compared to our method (of course this is testable). Notwithstanding the potential general usefulness of the work of Srivastava et al. (2016), this approach may be of questionable value when one is interested in fast and “accurate“ posterior inference, and moreover does not generalize to model variations such as the LCA, Levy Flights, etc.